



# Emission inventory of semi-volatile and intermediate volatility organic compounds and their effects on SOA over the Pearl River Delta region

Liqing Wu[1], Xuemei Wang[2,*], Sihua Lu[3], Min Shao[2], Zhenhao Ling[1,*]

[1] School of Atmospheric Sciences, Sun Yat-sen University, Guangzhou, 510275, China
[2] Institute for Environmental and Climate Research, Jinan University, Guangzhou, 510632, China
[3] College of Environmental Sciences and Engineering, Peking University, Beijing, 100871, China

*Correspondence to*: Prof. Xuemei Wang (eciwxm@jnu.edu.cn); Dr. Zhenhao Ling (lingzhh3@mail.sysu.edu.cn)

**Abstract.** Semi-volatile and intermediate volatility organic compounds (S/IVOCs) are considered as critical precursors of secondary organic aerosol (SOA), which is an important component of fine particulate matter ($PM_{2.5}$). However, the knowledge on the contributions of S/IVOCs to SOA is still poorly understood in the Pearl River Delta (PRD) region, southern China. Therefore, in this study, an emission inventory of S/IVOCs in the Pearl River Delta (PRD) region was developed for the first time for the year 2010. The S/IVOCs emission was calculated based on a parameterization method involving the emissions factors of POA (primary organic aerosol), emission ratios of S/IVOCs to POA, and domestic activity data. The total emission of S/IVOCs was estimated to be 323.4 Gg, with major emissions from central cities in PRD, *i.e.*, Guangzhou, Foshan, and Shenzhen. On-road mobile sources and industries were the two major contributors of S/IVOC emissions, with contributions of ~42% and ~35%, respectively. Furthermore, uncertainties of the emission inventory were evaluated through Monte Carlo simulation. The uncertainties ranged from -79% to 229%, which could be mainly attributed to mass fractions of OC (organic compound) to $PM_{2.5}$ from on-road mobile emissions and emission ratios of IVOCs/POA. The developed S/IVOC emission inventory was further incorporated into the Weather Research and Forecasting with Chemistry (WRF-Chem) model with a volatility basis-set (VBS) approach to improve the performance of SOA simulation and to evaluate the influence of S/IVOCs on SOA formation at a receptor site (Wan Qing Sha (WQS) site) of PRD. The following results could be obtained: (1) The model could resolve about 34% on average of observed SOA concentrations at WQS after considering the emissions of S/IVOCs, and 18%–77% with the uncertainties of the S/IVOC



emission inventory considered. (2) The simulated SOA over the PRD region was increased by 161% with the input of S/IVOC emissions, and it could be decreased to 126% after the reaction coefficient of S/IVOCs with OH radical was improved. (3) Among all anthropogenic sources of S/IVOCs, industrial emission was the most significant contributor of S/IVOCs for SOA formation, followed by on-road mobile, dust, biomass burning, residential, and off-road mobile emissions. Overall, this study firstly quantified emissions of S/IVOCs and evaluated their roles in SOA formation over PRD, which contribute towards significantly improving SOA simulation and better understanding of SOA formation mechanisms in PRD and other regions in China.

Keywords: S/IVOCs; emission inventory; uncertainties; SOA; WRF-Chem; PRD region

## 1 Introduction

As the key components, secondary organic aerosols (SOA) account for 20–80% of fine particulate matter ($PM_{2.5}$). They not only affect atmospheric chemistry, climate change, radiation balance, visibility, and air quality (Kanakidou et al., 2005; Pope et al., 2002), but also endanger human and vegetation health (Gehring et al., 2013; Zhou et al., 2014). In recent years, although $PM_{2.5}$ concentrations in major city clusters including the Pearl River Delta (PRD) region have shown a declining trend, the annual $PM_{2.5}$ concentrations are still higher than the World Health Organization (WHO) air quality standards and Air Quality Guideline (Li et al., 2015; Lin et al., 2018). Moreover, the contribution of SOA to $PM_{2.5}$ is increasing (Huang et al., 2015; Zheng et al., 2014), and it dominates the composition of $PM_{2.5}$ during episodes of photochemical smog. Therefore, investigating the formation mechanism of SOA is a prerequisite for better control over its precursors and $PM_{2.5}$, which is becoming increasingly more prominent as the concentrations of SOA precursors continue to increase over the years (Guo et al.et al., 2017).

Three-dimensional chemical transport models (CTM) have been widely used to investigate the formation and sources of SOA. Initially, organic compounds with similar properties or sources were clustered together in OA (organic aerosol) modules within gridded models due to the high complexity of OA and large varieties of compounds incorporated in the detailed chemical schemes (Johnson et al.,





2006), where large uncertainties occurred in simulations of SOA formation. Then a two-product model (Odum et al., 1996) based on the absorptive partitioning theory of Pankow (1994) and fitting methods developed from chamber study data was widely used in the simulation of SOA formation. However, this empirical two-product model was reported to largely under-predict SOA yield because it could not

account for the wide range of volatility of organic compounds. Recently, another scheme, *i.e..,* the volatility basis-set approach (VBS, typically one-dimensional VBS (1-D VBS)), was used to overcome the limitation of the two-product model (Donahue et al., 2006). In VBS, organic compounds are classified by their volatility, and it was developed on the basis of the absorptive partitioning theory. The VBS approach improves the modeling of further multigenerational oxidation processes and incorporates

low-volatility precursors of SOA, which consequently reduces the discrepancy between observation and simulation results. Furthermore, to capture the fragmentation process and oxidation of OA more accurately, the two dimensional VBS (2-D VBS) was proposed; 2-D VBS features a more detailed classification of organic compounds in different ranges of volatility and oxidation state. However, despite its potential for accurately simulating the evolution of SOA, it has rarely been used in CTM

because it involves much more complexity and computational expense compared to the widely used 1-D VBS (Donahue et al., 2011, 2012; Zhao et al., 2016a).

Although the scheme for SOA formation has been advanced significantly in recent years, large discrepancies have still been found between the observed and predicted abundance of SOA due to uncertainties in the formation mechanisms of SOA as well as the related parameterization (*i.e.,* SOA

yields of precursors of SOA) and the omittance of key precursors. For example, through recent chamber experiments, SOA yields of aromatics have been updated to be much higher than previous ones (Ng et al., 2007), while suggestions have also been made to consider wall losses of SOA in the SOA yields of each VOC precursor extracted from chamber experiments (Hildebrandt et al., 2009; Li et al., 2017a). In addition to the SOA yields, recent studies indicated that in-cloud aqueous-phase formation (Lim,

Carlton and Turpin, 2005; Ervens, Turpin and Weber, 2011) and oxidation of VOCs (volatility organic compounds) that were previously not considered in models (*i.e.*, isoprene, benzene and acetylene) could be important pathways for SOA formation (Claeys et al., 2004; Martín-Reviejo and Wirtz, 2005; Volkamer et al., 2009).



In addition to the traditional precursors (*i.e.*, VOCs), recent laboratory and modeling studies have suggested that semi-volatile and intermediate volatility organic compounds (S/IVOCs), which have effective saturation concentrations in the range of $10^{-2}$–$10^{3}\,\mu g/m^3$ and $10^{4}$–$10^{6}\,\mu g/m^3$ at 298 K and 1 atm, respectively, are key factors affecting the underestimation of SOA in numerical simulations (Donahue

et al., 2006; Robinson et al., 2007; Jiang et al., 2012). Therefore, to improve the performance of models simulating SOA formation, emissions and the chemical mechanism of S/IVOCs have been incorporated into different models, including box, regional, and global models (Robinson et al., 2007; Shrivastava et al., 2008; Grieshop, Donahue and Robinson, 2009; Pye and Seinfeld, 2010; Tsimpidi et al., 2010; Shrivastava et al., 2011; Ahmadov et al., 2012; Shrivastava et al., 2015; Woody et al., 2015). In terms of

chemical mechanisms, although 1-D VBS is still widely used in current models, one of the most important improvements is the adoption of the 2-D VBS scheme as mentioned above (Woody et al., 2015; Zhao et al., 2016a). For emission inventories, which is a prerequisite condition for improving model simulation of SOA formation and evaluating the roles of S/IVOCs in SOA production, previous studies typically estimated S/IVOC emissions from various sources based on the relationship of

S/IVOCs with POA, NMHCs (non-methane hydrocarbons) or naphthalene as well as emission profiles and source-specific volatility distribution factors for S/IVOC emissions extracted from various studies (Robinson et al., 2007; Pye et al., 2010; Shrivastava et al., 2011; May, Levin, et al., 2013; May, Presto, et al., 2013a, 2013b; Woody et al., 2015; Zhao et al., 2016a). However, very few studies have directly developed an emission inventory of S/IVOCs appropriate for CTM. For example, only Liu et al. (2017)

reported an emission inventory of vehicular IVOCs for China, estimating total emissions of IVOCs from vehicles in different provinces based on emission factors obtained from measurements of vehicle exhaust in the United States. However, this emission inventory was flawed as it could not be applied to CTM because the total IVOC emissions had not been spatially allocated into grid cells, and it was not sufficiently localized as the emission factors of IVOCs from vehicle exhaust were completely based on

measurements in the United States (Zhao et al., 2015, 2016b). As such, most modeling studies used the same parameterizations and volatility distributions of all emissions independent of source types to simulate SOA formation. However, S/IVOC emissions and characteristics of SOA and particles widely vary among different countries and regions.




The significance of S/IVOCs have been demonstrated through lab and modeling studies in different environments, but the emissions of S/IVOCs and their roles in the formation of SOA in China is still poorly understood, especially in the PRD region, where photochemical smog and high oxidative capacity are frequently observed (Hofzumahaus et al., 2009; Xue et al., 2016). Therefore, in the present study, a gridded emission inventory of S/IVOCs for the PRD region was first developed and then incorporated into the WRF-Chem model (Weather Research and Forecasting model with Chemistry) with the 1-D VBS approach. The objectives of this study are as follows: (i) to examine the potential of considering S/IVOCs for improving the simulation of SOA formation; (ii) to evaluate the contributions of S/IVOC to SOA over the PRD region. This study is the first report focusing on the emissions of S/IVOCs and their contributions to SOA formation in the PRD region, which could advance the understanding of SOA formation mechanisms in PRD and could be extended to other regions in China.

## 2 Methodology

### 2.1 Establishment of the S/IVOC emission inventory

In this study, a gridded emission inventory of S/IVOCs was determined by using Eq. (1).

$$E_{S/IVOCs,j} = \sum_{j,k} A_{j,k} \times EF_{S/IVOCs,j} \times (1-\mu) \times 10^{-3} \qquad (1)$$

The detailed description for the definition of parameters has been provided in section S1 of the supplementary material. Emission factors of S/IVOCs were calculated on the basis of existing traditional POA emission factors using source-specific linear scaling factors because available emission factors of S/IVOCs are limited. Moreover, the traditional POA emission factors of different sources were obtained from POC (primary organic carbon) emission factors using source-specific ratios of OM/OC (ratios of organic mass to organic carbon), while the POC emission factors were obtained from $PM_{2.5}$ emission factors by applying the source-specific mass fractions of OC to $PM_{2.5}$. Therefore, Eq. (1) was extended to Eq. (2). The related parameters of S/IVOCs, including the activity levels, removal efficiency and spatiotemporal allocations, were assumed to be the same as those of POA and POC for all source categories. S/IVOC emissions in the PRD region for the year 2010 was calculated using Eq. (3), among which the activity levels, removal efficiency, and emission factors were combined and



expressed as $PM_{2.5}$ emissions ($E_{PM2.5,j}$ in Eq. 3). Note that the $PM_{2.5}$ emissions in this study was obtained from a highly resolved spatial anthropogenic PRD regional emission inventory for the year 2010 with a horizontal resolution of 3 km (Zheng et al., 2010b).

$$E_{S/IVOCs,j}= \sum_{j,k} A_{j,k} \times EF_{PM_{2.5},j} \times F_{OC,j} \times \frac{OM}{OC_j} \times (\frac{E_{SVOCs,j}}{E_{POA,j}} + \frac{E_{IVOCs,j}}{E_{POA,j}}) \times (1-\mu) \times 10^{-3} \quad (2)$$

$$E_{S/IVOCs,j}= E_{PM_{2.5},j} \times F_{OC,j} \times \frac{OM}{OC_j} \times (\frac{E_{SVOCs,j}}{E_{POA,j}} + \frac{E_{IVOCs,j}}{E_{POA,j}}) \quad (3)$$

The above parameters used in the development of the emission inventories of S/IVOCs of each source category were extracted from recent studies (Table 1). In order to calculate the oxygen fraction and ratios of the non-oxygen component to the carbon component of each species in all anthropogenic sources, which would be required in modeling (section 2.2.2), elemental ratios of O/C, H/C and N/C ratios were also estimated.

Furthermore, uncertainties of the emission inventory of S/IVOCs, which can be attributed to uncertainties in all parameters, were evaluated and quantified using statistical methods and Monte Carlo simulation, as suggested by Zheng et al. (2010a). Sample correlation coefficients between total S/IVOCs emission and model input parameters or S/IVOCs emission of each specific source category have been calculated to identify the key sources of uncertainties in the estimation of S/IVOC emissions (NARSTO, 2005). Based on the different values of model input parameters from previous studies (Table 1), probabilistic distributions representing uncertainty ranges of different parameters, including $F_{OC}$, $E_{SVOCs}/E_{POA}$, $E_{IVOCs}/E_{POA}$, OM/OC, O/C, H/C, and N/C, from different source categories are summarized in Table 2. Additionally, uniform distribution based on the results of uncertainty assessment in Zhong et al. (2018) was applied to all source categories of $PM_{2.5}$ emission in the present study.

## 2.2 Model description and settings

### 2.2.1 Model settings

To further evaluate the roles of S/IVOCs in SOA formation, the newly developed S/IVOC emission inventory was incorporated into the WRF-Chem model to simulate the formation of SOA and investigate its impact factors in the PRD region. The WRF-Chem model (https://ruc.noaa.gov/wrf/wrf-





chem/) is a fully coupled online meteorology–chemistry model that can be used to simulate physical and chemical processes simultaneously (Grell et al., 2005). This model has been widely used to simulate the formation of secondary products (*i.e.*, SOA and $O_3$), including their relationship with precursors, the influence of meteorological conditions, and contributions from anthropogenic and biogenic emissions

from regional to cloud resolving scales (Fast et al., 2009, 2006; He et al., 2015; Jiang et al., 2012; Li et al., 2011; Liu, 2014; Sharma et al., 2017; Tie et al., 2013).

The model configuration is presented in Table 3, and the model domain is presented in Fig. 1. The simulation was conducted from 1200 UTC on November 17, 2008 to 0000 UTC on November 26, 2008 because measured data of SOA were available from November 19 to 25, 2008 at the receptor site of the

PRD region, *i.e.*, the Wan Qing Sha (WQS) site. During the simulation period, the first 24 h were consumed as the spin-up time for the simulation. The initial meteorological field and boundary meteorological conditions were provided by the ERA-Interim reanalysis dataset from the European Centre for Medium-Range Weather Forecasts (ECMWF) with the resolution of $0.5° \times 0.5°$, while the chemical boundary condition was obtained from the Model for Ozone and Related chemical Tracers

(MOZART) global simulation of trace gases and aerosols (Emmons et al., 2010). The above initial field and boundary meteorological conditions have been confirmed to be appropriate for the reproduction of observed meteorological parameters in PRD (Situ et al., 2013).

The gas-phase chemistry mechanism used in the simulation was SAPRC-99 (Statewide Air Pollution Research Center), including 235 reactions of 80 gases (Carter, 1999). It should be noted that the gas-

phase photochemical oxidation of gas-phase organic species for the formation of SOA, *e.g.*, the gas-phase chemistry of BVOCs (biogenic VOCs) such as isoprene, monoterpenes and sesquiterpenes, has been recently updated and incorporated in the mechanism (Situ et al., 2013). In addition, the Model For Simulating Aerosol Interaction and Chemistry (MOSAIC) aerosol chemistry mechanism coupled with 2-species VBS treatment was used to represent aerosol processes (Zaveri et al., 2008). The MOSAIC

scheme includes aerosol species, such as sulfates, nitrates, ammonium salts, sodium salts, chlorine salts, calcium salts and other inorganics (OIN), organic carbon (OC), elemental carbon (EC), and water, but does not consider the formation of SOA from organic vapors (Fast et al., 2009). Therefore, this



mechanism was coupled with a simplified two-species 1D-VBS (section 2.2.2) developed by Shrivastava et al. (2011) to simulate the formation of OA.

Anthropogenic emissions of $PM_{10}$, $PM_{2.5}$, VOCs, NOx, $SO_2$, and CO in the PRD region were derived from a highly resolved spatial anthropogenic PRD regional emission inventory for the year 2010 with a

horizontal resolution of 3 km, whereas emissions outside the PRD region were based on the Guangdong provincial emission inventory ( Zheng et al., 2010b). This emission inventory was developed using the best available domestic emission factors and activity data, including the sectors of industry, on-road and off-road mobile sources, residential sources, dust, and biomass burning (Zheng et al., 2009). BVOC emissions were derived from the Model of Emissions of Gases and Aerosols from Nature (MEGAN,

https://sites.google.com/uci.edu/bai) developed by Guenther et al. ( 2012).

**2.2.2 VBS approach**

With the configuration mentioned above, the WRF-Chem model used in this study provides a simplified and computationally efficient 2-species 1D-VBS scheme coupled with MOSAIC that includes V-SOA formed by the oxidation of VOCs, traditional SOA precursors emitted from varied anthropogenic and

biogenic sources, and SI-SOA formed by the oxidation of S/IVOCs—untraditional SOA precursors emitted from anthropogenic sources. This scheme was simplified from the detailed 9-species VBS, a scheme with more surrogate organic compounds categorized by different ranges of volatility (Shrivastava et al., 2011). The simplified 2-species scheme categorized S/IVOCs into two volatility species with effective saturation concentration C* equal to 0.01 and $10^5$ μg/m³ at 298K and 1 atm. C* is

the inverse of the Pankow-type equilibrium partitioning coefficient, which describes the fraction of gas and particle components in SOA formation. Note that gas phase SVOCs and all IVOCs in this mechanism are represented by species with C* equal to $10^5$ μg/m³, and IVOCs were considered to remain in the gas phase before photochemical oxidation in the atmosphere. POA and SI-SOA were assumed to be non-volatile in the model. POA is the remaining aerosol component after the evaporation

of gas phase SVOCs. In this mechanism, POA with SI-SOA are represented by species with C* equal to 0.01 μg/m³. This simplified 2-species VBS has been confirmed to be more suitable for computationally extensive models (*i.e.*, WRF-Chem) for running complex coupled cloud-aerosol-meteorology because




of its similar predictions of total OA mass, individual OA components, and evolution of organic aerosols in addition to its reduction in computational cost by a factor of 2, as compared to the detailed 9-species VBS (Shrivastava et al., 2011).

In terms of the formation of SI-SOA in the 2-species VBS, the primary oxidation of S/IVOCs

transformed the gas-phase high-volatility S/IVOCs (C*=$10^5$ µg/m³) into the extremely low-volatility SI-SOAs (C*=0.01 µg/m³) with the OH reaction rate constant ($k_{OH}$) of $0.57 \times 10^{-11}$ cm³ molecule⁻¹ s⁻¹ and oxygen yield of 50%. In order to align the SOA predictions between 2-species and 9-species VBS schemes, $k_{OH}$ in the 2-species VBS had been reduced by a factor of 7 (*i.e.*, $0.57 \times 10^{-11}$ cm³ molecule⁻¹ s⁻¹) from that of the 9-species VBS ($4 \times 10^{-11}$ cm³ molecule⁻¹ s⁻¹) because the orders of magnitude

reduction in volatility through one generation of oxidation was 7 times that in the 9-species VBS. Note that the $k_{OH}$ of $4 \times 10^{-11}$ cm³ molecule⁻¹ s⁻¹ in the 9-species VBS was assumed to be perhaps 50% higher than that of a typical large saturated n-alkane (Atkinson and Arey, 2003; Robinson et al., 2007). The specific oxidation reaction equations are as follows (the detailed description for the parameters in the equations was provided in the supplementary material):

$$\text{S/IVOC(g)}_{2,e,c} + \text{OH} \rightarrow \text{SI-SOA(g)}_{1,e,c} + 0.5 \text{ SI-SOA(g)}_{1,e,o} \qquad (4)$$

            $$\text{S/IVOC(g)}_{2,e,o} + \text{OH} \rightarrow \text{SI-SOA(g)}_{1,e,o} + \text{OH} \qquad (5)$$

In addition, V-SOA formed by the oxidation of anthropogenic and biogenic VOCs in this mechanism was considered using one-species treatment with the configuration of the saturation concentration C* of V-SOA as 1 µg/m³ at 298K and 1 atm. NO$_x$ dependent fixed 1-product yields for all VOCs precursors

were proposed by Shrivastava et al. (2011).

Gas-particle partitioning between the gas and aerosol phases of both SI-SOA and V-SOA was calculated using the absorptive partitioning theory as described by Donahue et al. (2006):

$$C_{i,a} = \frac{C_{i,tot}}{1 + C_i^*/M} \qquad (6)$$

where $C_{i,a}$ denotes aerosol phase SOA mass concentration at a given volatility bin $i$ (here, its bin

boundaries are C* values of 0.01 and/or $10^5$ µg/m³); $C_{i,tot}$ denotes the total mass concentrations of gas- and aerosol-phase SOA for bin $i$; $C_i^*$ denotes the saturation concentration for bin $i$; M denotes total mass concentrations of OA, which includes POA and SOA.





To calculate the influence of temperature on $C^*$, the Clausius-Clapeyron equation was used:

$$C_i^* = C_{i,0}^* \frac{T_0}{T} \exp\left(\frac{\Delta H_i}{R}\left(\frac{1}{T_0} - \frac{1}{T}\right)\right) \qquad (7)$$

where $C_i^*$ and $C_{i,0}^*$ denotes saturation concentration at T and $T_0$ (reference temperature 298K), respectively, for bin $i$; R is the universal gas constant; $\Delta H_i$ denotes the enthalpy of vaporization for bin $i$.

### 2.2.3 Model scenarios

In order to evaluate the roles of S/IVOCs on the formation of SOA over the PRD region, thirteen simulations were performed from November 19 to 25, 2008, including one control BASE simulation and twelve sensitivity CASE simulations. Table 6 provides detailed description on the base and sensitivity scenarios. For the base scenario, the simulation was conducted without the input of S/IVOC emissions. For CASE1, this simulation was conducted with the input of S/IVOCs from all anthropogenic emissions (section 3) in order to estimate the contributions of S/IVOCs to the formation of SOA.

A large uncertainty of $0.57 \times 10^{-11}$ cm$^3$ molecule$^{-1}$ s$^{-1}$ was found for the $k_{OH}$ of S/IVOCs species in the 2-species VBS used in current WRF-Chem model, which was calculated on the basis of the $k_{OH}$ of the 9-species VBS that was assumed to be about 50% higher than that of a typical large saturated n-alkane (Atkinson and Arey, 2003; Robinson et al., 2007). In this study, the $k_{OH}$ of S/IVOCs species was updated according to the emission factors and $k_{OH}$ of 57 speciated IVOCs from the vehicular emission measurements (Zhao et al., 2015, 2016b) using the molar weighting method by the following equation (Carter, 1999):

$$k_{OH} = k_{OH,i} \times \frac{EF_i}{EF_{tot}} \qquad (8)$$

where $i$ denotes the specific S/IVOCs species; $tot$ denotes all S/IVOCs species; EF denotes the emission factor; $k_{OH}$ denotes the OH reaction rate constant. The $k_{OH}$ of S/IVOCs was calculated to be $3 \times 10^{-11}$ cm$^3$ molecule$^{-1}$ s$^{-1}$, which is smaller than the original $k_{OH}$ of $4 \times 10^{-11}$ cm$^3$ molecule$^{-1}$ s$^{-1}$ in the model. Then, the reaction rate with OH radicals was reduced to $0.42 \times 10^{-11}$ cm$^3$ molecule$^{-1}$ s$^{-1}$ by a factor of 7 in order to ensure its applicability to the 2-species VBS scheme, as suggested in section 2.2.2. To





evaluate the effect of the OH reactivity of S/IVOCs on the formation of SOA, CASE2 was conducted using the new updated $k_{OH}$ with the input of the same S/IVOC emission as in CASE1.

For CASE3-6, the simulations were designed with varied amounts of S/IVOC emissions at the 50% and 95% confidence intervals (section 3.2) using the new updated $k_{OH}$ of S/IVOCs in order to evaluate the

sensitivity of the SOA simulation to the magnitude of S/IVOC emissions and quantify the uncertainty ranges in SOA prediction attributable to uncertainties of S/IVOC emissions. Note that CASE3 and CASE6 were conducted with the lower and upper limits of the uncertainty ranges of S/IVOC emissions estimated at the 95% confidence interval (which was 21% and 329% of the amounts in inventory as suggested in section 3.2) as presented in Table 5, whereas CASE4 and CASE5 were conducted with the

edges of the uncertainty ranges of S/IVOC emissions estimated at the 50% confidence interval (45% and 127% of the amounts in the inventory as suggested in section 3.2). Furthermore, CASES7-12 were simulated using the new updated $k_{OH}$ of S/IVOCs with only the input of individual S/IVOC emissions, *i.e.*, biomass burning, dust, industry, off-road mobile, on-road mobile and residential sources, to quantify the contributions of each S/IVOC emission to the formation of SOA.

## 3 Emission inventory of S/IVOCs

### 3.1 S/IVOC emissions

Using the parameterization method described in Eq. (3), hourly gridded S/IVOC emissions in the PRD region for the year 2010 with a resolution of 3 km × 3 km were estimated with parameters given in Table 2 and the high-resolution $PM_{2.5}$ emission inventory (Zheng et al., 2010b). As shown in Table 4,

the total S/IVOC emission in the PRD region is 323.4 Gg in 2010, of which on-road mobile sources contributes about 41.6% (134.4 Gg), industry about 35.4% (114.6 Gg), dust about 14.5% (46.8 Gg), biomass burning about 4.5% (14.4 Gg), residential sources about 2.6% (8.4 Gg), and off-road mobile sources about 1.5% (4.8 Gg). Regarding city-level contributions, Guangzhou was the largest contributor to S/IVOC emissions with a contribution of 23.9%, followed by Foshan (18.4%), Shenzhen (15.1%),

Jiangmen (11.9%), and Dongguan (11.7%) as shown in Fig. 2. Notably, as expected, on-road mobile sources and industry, which involve large amounts of vehicles and industrial plants, were the top two



contributors in all cities, except for Zhongshan, where the contribution of dust to total S/IVOC emissions was higher than industry because of the accelerating urbanization with frequent urban constructions but much less industrial plants than in Guangzhou, Foshan, Dongguan, and Shenzhen (Pan et al., 2015; Yin et al., 2015; GSY, 2010).

Figure 3a-f show the spatial distributions of S/IVOCs emitted from different sectors for the year 2010. In general, the spatial characteristics of S/IVOC emissions in 2010 were consistent with the distribution of on-road mobile and industrial emissions (Fig. 4), the top two S/IVOC contributors in this region. Furthermore, the spatial distributions of total S/IVOC emissions agreed well with the road network with the high S/IVOC emissions located in central cities including Guangzhou, Foshan, Dongguan, and

Shenzhen. The high industrial emissions of S/IVOCs were mainly concentrated in Foshan, Dongguan, Zhongshan, and Guangzhou, where large amounts of industrial point sources and power plants exist (Fig. 3c). In contrast, large amounts of emissions from biomass burning were found in Zhaoqing, Jiangmen, and Huizhou, which are characterized by extensive combustion of household firewood and straw associated with the large rural populations (Fig. 3a), contributing nearly 43% to total rural

populations in the PRD region in 2010 (GSY, 2011; Yuan et al., 2010). The spatial distributions of S/IVOCs emitted from on-road mobile sources were very consistent with the patterns of road networks in the PRD region. The emissions were concentrated in central economically developed cities with large numbers of vehicles (Fig. 3e). High S/IVOC emissions from dust were mainly concentrated in Guangzhou, Foshan, Dongguan, Shenzhen, and Zhongshan, associated with the heavy traffic flows and

frequent urban constructions because of the preparation of the 2010 Asian Games and the accelerating urbanization processes in recent years (Fig. 3b). Compared with the abovementioned sectors, S/IVOC emissions from residential and off-road mobile sources in the PRD region were lower (Fig. 3d and f). Nevertheless, the total S/IVOC emission (323.4 Gg) was only a quarter of total VOC emission (1224.5 Gg) in the PRD region in 2010 (Fig. 5), but it was more than 6 times the total OC emission (52.9 Gg).

Moreover, the contributions of different sectors varied in different emission inventories. For example, biogenic and solvent use sources totally contributed to the overall VOC emissions by 45% but did not contribute to emissions of S/IVOCs, $PM_{2.5}$, and OC. The contribution of biomass burning (4%, 14.4 Gg) to S/IVOCs was much smaller than that to OC (24%, 12.9 Gg) because the emission ratio of IVOCs to



POA for biomass burning is much smaller than that of other sectors. Industrial sources contributed less to S/IVOC emissions than to $PM_{2.5}$ with contributions of 35% and 52%, respectively, while on-road mobile contributed more to S/IVOC emissions (42%, 134.4 Gg) as the fraction of OC in $PM_{2.5}$ ($F_{OC}$) in on-road mobile emissions was higher than that in industrial emissions, when other parameters in the emission model for these two sectors were similar (Table 2).

## 3.2 Uncertainties in S/IVOC emissions

An uncertainty assessment of the 2010 PRD regional S/IVOC emission inventory together with a sensitivity analysis based on the Monte Carlo simulation and sample correlation coefficient method (Zheng et al., 2010a), were performed to determine the ranges of uncertainties and identify the key sources of uncertainties in S/IVOC emission estimates. Table 5 lists estimated ranges of uncertainties and associated correlation coefficients with estimated total S/IVOC emissions for the different parameters used in calculating the total S/IVOC emissions in different source categories. As shown in Table 5, the uncertainty in the total S/IVOC emissions was very high with a relative error of -79–229% at the 95% confidence interval, which could be mainly attributed to uncertainties in the S/IVOC emissions of the on-road mobile sources because of the largest correlation coefficient of the on-road mobile S/IVOC emissions with total S/IVOC emissions among all the source categories. It is noteworthy that the uncertainty ranges of the emission inventories of S/IVOCs were wider than those of VOCs and $PM_{2.5}$, which were only -6–99% and -6–77%, respectively (Zhong et al., 2018). For input parameters in the emission model, the correlation coefficients between total S/IVOC emissions and $F_{OC}$ for the on-road mobile sources or ratios of $E_{IVOCs}/E_{POA}$ for all source categories, except biomass burning, were very large, indicating that these parameters were the key sources of high uncertainties in the S/IVOC emission estimates. It should be noted that the actual uncertainties in S/IVOC emission estimates should be larger because the same ratios of $E_{IVOCs}/E_{POA}$ and $E_{SVOCs}/E_{POA}$ were used for all source categories, except biomass burning, which were only based on measurements of vehicular emission. These results indicated that more measurement of $F_{OC}$ from on-road mobile emission and source-specific measurements of $E_{IVOCs}/E_{POA}$ and $E_{SVOCs}/E_{POA}$ is key to reducing uncertainties in S/IVOC emission estimates.




To indicate the difference of S/IVOC emission inventory developed by different methods, the S/IVOC emission inventory developed in this study was further preliminarily compared to recently proposed global emissions of PAHs (Polycyclic Aromatic Hydrocarbons) by Shen et al. (2013) (the detailed data for the emission inventory was provided in Table S1 in the supplementary material) because PAHs are

5 important components of S/IVOCs. In this study, emissions of five PAHs including NAP (naphthalene), ACY (acenaphthylene), FLO (fluorene), PHE (phenanthrene), and PYR (pyrene) with high fractions in total PAH emissions were selected for comparison with corresponding PAH emissions of Shen et al. (2013) for the year 2010 in the PRD region. Note that the emission of individual PAHs in all source categories in this study was extracted from the total IVOC emission using the ratio of specific

individual PAH to total IVOCs from vehicular emission measurements (Zhao et al., 2015, 2016b). Large deviations were found for emissions of the abovementioned PAHs between the present and previous studies, especially for NAP (Fig. S1). For example, NAP emissions were larger with more distinct spatial characteristics in this study than in Shen's inventory over most of the PRD region. The characteristic of road networks was also observed for the spatial distribution of NAP emission in the

present study, which was not reflected in Shen's inventory. Furthermore, the total emissions of the 5 selected PAHs over PRD in this study were about 3.4 times those in Shen's inventory, with multiples ranging from 1.2 to 13.4 in nine individual cities of PRD (Table S1). The discrepancies in PAH emissions in different studies can be mainly attributed to the following factors: 1) Differences in the resolution of emission inventories. For example, the spatial resolution of the emission inventory in this

study was 3 km × 3 km, which is much higher than that of the previous study (0.1° × 0.1°). 2) Differences in the parameterization methods for developing different inventories. For example, emission factors of PAHs in the present study were calculated on the basis of those of IVOCs using the ratio of specific individual PAH to total IVOCs, wherein the emission factors of IVOCs were obtained from those of existing traditional POA using source-specific linear scaling factors. However, emission

factors of PAHs in Shen's study were directly obtained from actual measurements from various reports, which were further calculated to be time-specific based on the regression model and technology splitting approach. Nevertheless, by considering the uncertainties of different inventories (*i.e.,* -55–27% and -34–62% at the 50% confidence interval for emission inventories of the present and previous



studies, respectively), it is reasonable to conclude that the emissions of selected PAHs between the two studies are comparable. Moreover, further investigation revealed that the spatial variations of PAH emissions in this study may be more reasonable than those in Shen's inventory. For example, high centers of PAH emissions in Shen's inventory were located only in Guangzhou and Shenzhen. On the

5 other hand, in this study, high PAH emissions were found in central cities including Guangzhou, Foshan, Shenzhen, and Dongguan, which have dense traffic and population. This result is consistent with the result that traffic was frequently found to be one of the most important sources of PAH emissions (Riva et al., 2017).

## 4 The simulation results

### 4.1 Effects of S/IVOCs on SOA concentration

In this study, daily measured concentrations of SOA at the WQS site in Guangzhou were used to evaluate the model performance on the simulation of SOA (Ding et al., 2012). The monitoring data of this site could represent the regional background of air pollution in the PRD because it is surrounded by large farmland and rare traffic with flat terrain (Ding et al., 2012). The time series of SOA for BASE,

CASE1, CASE2, and observation during the study period is plotted in Fig. 6a. Both BASE and CASE simulations well reproduced the day-to-day variations of SOA, although the model could not capture the observed high concentrations of SOA. Another remarkable feature in Fig. 6a is that the predicted concentrations of SOA became much closer to the observed values after the S/IVOC emissions were considered, with the discrepancy between simulations and observations decreasing from -9.15 μg/m$^3$ to

20 -6.39 μg/m$^3$ for CASE1 (30% decrease, $p < 0.01$). Moreover, the performance of SOA simulation was improved by 196% for CASE1 compared to BASE. The ratios of predicted SOA to observed SOA in CASE1-2 and BASE runs are presented in Fig. 6b. The model could resolve 39% of the observed SOA with an increase of 26% as compared to BASE when the S/IVOC emissions were included and the original $k_{OH}$ of S/IVOCs was used. Figure 7a shows the relative variations of SOA between CASE1 and

25 BASE in the whole modeling domain. An obvious increase of 40–375% of SOA is found over the PRD region with an average regional increase of 161% when S/IVOC emissions were incorporated into the





model. The most remarkable increase patterns are found in the cities of Foshan, Shenzhen, Dongguan, and Jiangmen, with the increment ranging from 240% to 375%. This is consistent with the spatial SI-SOA in Fig. 8 and can be probably attributed to the high anthropogenic S/IVOC emissions in these cities (Fig. 4). Furthermore, a substantial increase of SOA was found in the southwest downwind area of

the PRD region with increments of 240–325% attributable to the influences of both local pollutants and pollutants transported from the upwind area because the dominant wind direction over the PRD region was northeasterly during the pollution period (Fig. 8). Notably, high increasing ratios of SOA concentrations in Guangzhou only appeared in a small southwestern part of Guangzhou, probably because high S/IVOC emissions in Guangzhou mainly resulted in considerable SOA growth in

downwind areas, especially Foshan, which lies to the southwest of Guangzhou. Nevertheless, the above results demonstrated that S/IVOCs are significant precursors for forming SOA, and the model performance on SOA formation could be improved significantly if S/IVOCs emissions were considered. Therefore, besides traditional VOCs, S/IVOC emissions should be included in CTM to achieve accurate modeling of the formation of SOA and regional air quality.

In contrast, the predicted SOA in CASE2 decreased by 14% as compared to CASE1 after the $k_{OH}$ of S/IVOCs was improved. Moreover, the model could resolve 34% of the observed SOA at the WQS site, which is smaller than the resolved fraction of 39% in CASE1 (Fig. 6a-b). The average regional increase ratio of SOA decreased to 126% in CASE2 with the newly updated decreased $k_{OH}$ of S/IVOCs and the same S/IVOC emissions as in CASE1 (Fig. 7b). This suggests that the decreased OH reactivity

coefficient indeed decreased the formation rate of SOA, and a more precise OH reactivity is required for the model to better simulate SOA.

CASE3-6 were simulated with the input of varied amounts of S/IVOC emissions on the basis of the uncertainty ranges of the estimates of S/IVOC emissions (Table 6 and section 2.2.3). The uncertainty ranges of the ratios of predicted SOA concentrations to observed ones, attributable to uncertainties in

S/IVOC emissions, are presented as an error bar in Fig. 6b. As expected, the ratios of temporal average simulated SOA to observed SOA at WQS site during the study period varied from 18% to 77% after taking the uncertainties of S/IVOC emissions into account. Figure 7c-d show minor increases of SOA with the input of lower S/IVOC emissions for CASE3 and CASE4, as compared to CASE1, with




average regional increases of 27% and 57%, respectively. Figure 7e-f show larger increases of SOA with the input of higher S/IVOC emissions for CASE5 and CASE6, with average regional increase of 158% and 395%, respectively. The results suggest that SOA is strongly sensitive to the amounts of S/IVOC emissions. Consequently, it is of great importance to reduce the uncertainties in the S/IVOC

emission inventory to achieve accurate simulations of SOA.

## 4.2 Key anthropogenic S/IVOCs for SOA formation

Six simulations including CASE7-12 were conducted with only the input of S/IVOC emissions from individual source categories in order to identify the key anthropogenic source of S/IVOCs to form SOA, as described in Table 6. The spatial distributions of the relative differences of predicted SOA

concentrations between CASE simulations and BASE are presented in Fig. 9. The increasing ratio of SOA in CASE9 was found to vary in the range of 5–190% over the PRD region with an average increase of 52% when the industrial S/IVOC emission was incorporated into the model. The center of increasing SOA was located in Foshan, which is an industrially developed city (Fig. 9c). After including on-road mobile S/IVOC emissions into the model, the predicted SOA was increased by 5–180% with an

average regional increase of 43% (Fig. 9e), and high amounts were detected over central cities including Shenzhen, Guangzhou, Foshan, and Zhongshan, which feature a high rate of vehicle ownership, contributing to 71% of total vehicle ownership in the PRD region (GSY, 2011). After considering S/IVOCs emitted from dust, the average regional increase ratio of SOA was 18% (Fig. 9b), and the centers were located in Guangzhou, Foshan, Zhongshan, Dongguan and Shenzhen. These cities have

high traffic flows and frequent urban constructions, and their vehicle ownerships and floor space of buildings under construction contributed to ~86% and ~81% of those in the PRD region (Pan et al., 2015; GSY, 2011). With the input of S/IVOC emissions from biomass burning, the average regional increasing ratio of SOA was up to 8% (Fig. 9a), and high values were mainly distributed in Zhaoqing. This city has expansive agricultural areas and large rural populations, accounting for ~31% and ~15%,

respectively, of the total in the PRD (Yang et al., 2013; Pan et al., 2015; GSY, 2011). Nevertheless, the average regional SOA increased by only 2% and 4% with the input of S/IVOCs emitted from off-road mobile and residential sources, respectively (Fig. 9d and f). Notably, similar high centers of increasing





SOA and S/IVOC emissions could be found in Fig. 9 and Fig. 3, respectively, for six specific sectors, indicating that the increment in SOA concentrations was highly correlated with the input of S/IVOC emissions. Overall, the industry and on-road mobile sources were the main anthropogenic sources of S/IVOCs contributing to the formation of SOA in the PRD region, followed by dust, biomass burning,

residential, and off-road mobile sources. However, it was of interest to find that though the emission strength of on-road mobile S/IVOCs was stronger than that of industrial S/IVOCs in PRD, the contributions of industry to SOA formation was higher than on-road mobile sources. This is related to different transport patterns in varied simulations with the input of S/IVOC emissions from different source categories. For example, high industrial S/IVOC emissions outside PRD would induce

considerable SOA growth downwind inside PRD; however, high on-road mobile S/IVOC emissions in coastal cities such as Shenzhen would bring the SOA growth to the South China Sea, resulting in a loss of SOA inside PRD.

## 5 Discussion and conclusions

In this study, a highly resolved gridded emission inventory of S/IVOCs for the PRD region in 2010 was

developed. The estimates showed that total S/IVOC emission in the PRD region for the year 2010 was 323.4 Gg, 77% of which could be attributed to on-road mobile and industrial sources. Large uncertainties were still observed in S/IVOC emission estimates, with relative error ranging from -79% to 229%. These uncertainties could be attributed to the $F_{OC}$ of the on-road mobile source and $E_{IVOCs}/E_{POA}$ ratio of all source categories, except biomass burning. Therefore, these parameters should

be prioritized in further experimental studies in order to improve future S/IVOC emission inventories. Moreover, thirteen simulations using the WRF-Chem model were conducted to investigate the effects of S/IVOCs on SOA and identify the key anthropogenic source of S/IVOCs contributing to SOA formation over the PRD region. The analysis of the simulation results indicated that the performance of SOA simulation was greatly improved after considering the reaction pathway producing SI-SOA from

S/IVOCs. S/IVOCs could result in considerable SOA growth, and the $k_{OH}$ of S/IVOCs had a non-negligible effect on the production of SI-SOA. After considering the uncertainties of S/IVOC emissions, the model could resolve 18%–77% of observed SOA concentrations at WQS site. These indicate the





need for more experimental data of $k_{OH}$ for S/IVOCs to reduce the uncertainties of this parameter within the model, and reduction of uncertainties of S/IVOC emissions to more accurate simulation of SOA formation. In addition, the industrial and on-road mobile sources were the top two important anthropogenic sources of S/IVOCs contributing to SOA formation, followed by dust, biomass burning,

residential, and off-road mobile sources.

Although the performance of the model in simulating SOA could be significantly improved, many issues still remain to be resolved. The observed SOA concentrations could not be accurately reproduced in the present study, especially for high SOA concentrations (Fig. 6a). We inferred that the incomplete and inaccurate formation mechanism of SOA, and large unresolved uncertainties in the S/IVOC

emission inventory were the main reasons for the underestimation of SOA concentrations in the simulation. For example, large uncertainties still remained within source-specific and season-specific S/IVOC emissions, reaction rates of S/IVOCs, and SOA yields of VOCs and S/IVOCs. Furthermore, specific profiles of S/IVOCs were lacking. The approach including the distribution of S/IVOC emissions was based on inadequate data from domestic and foreign studies without sufficient

localization in the PRD region, which further included large uncertainties. Furthermore, the assumption that SVOC emissions were included in POA emissions was not sufficiently constrained because of the limited observation data of HOA and BBOA. Therefore, much more local experimental work is needed to quantify all the abovementioned parameters in the future. In addition, the introduction of complete and complex physical and chemical processes of SOA formation, *e.g.*, gas- and aqueous-phase

oxidation, heterogeneous and accretion reactions, acid catalysis reactions of SOA from glyoxal, and chemical aging of SOA, may be useful in estimating SOA concentrations more accurately although it will increase experimentation costs and introduce larger uncertainties (Carlton et al., 2008; Denkenberger et al., 2007; George and Abbatt, 2010; Hallquist et al., 2009; Kroll and Seinfeld, 2008; Liggio et al., 2005; Pun and Seigneur, 2007; Washenfelder et al., 2011). For example, Dzepina et al.

(2011) found that including chemical aging of V-SOA resulted in larger regional overprediction of SOA, whereas Ahmadov et al. (2012) reported a good agreement with observations after considering it. Shrivastava et al. (2011) pointed out that aging parameterization based on smog chamber measurements involves large uncertainties because the time-scales of photochemical ages are longer than the one



accessible in chambers. Shrivastava et al. (2015) also pointed out that neglecting fragmentation reactions in aging parameterizations leads to large model overpredictions of SOA concentrations at all surface sites. Therefore, we plan to test more chemical processes that have not yet been considered in the WRF-Chem model and introduce the parameters required for establishing the S/IVOC emission

inventory and model parameterization with less uncertainties based on more local experimental work in the future. Furthermore, we plan to build the S/IVOC emission inventory based on ample local directly measured S/IVOC emission factors and volatility distribution factors of POA in the future work instead of scaling POA emission factors.

*Data availability.* The underlying research data in this study can be accessed by contacting the corresponding author.

*Author contribution.* In this study, the analysis methods were developed and the whole structure for the manuscript was designed by Dr. Zhenhao Ling and Prof. Xuemei Wang. Miss Liqing Wu conducted the

data process and wrote the original copy of the manuscript. Miss Sihua Lu and Prof. Min Shao provided the related data and made revision on the manuscript. Furthermore, the simulations using the WRF-Chem model were designed and conducted by Miss Liqing Wu. Finally, the manuscript was finalized by Dr. Zhenhao Ling and Prof. Xuemei Wang.

*Competing interests.* The authors declare that they have no conflict of interest.

*Acknowledgements.* This work was supported by the State Key Program of National Natural Science Foundation of China (No. 91644215), the National Key Research and Development Program of China (2017YFC0210106 and 2016YFC0202206) and National Nature Science Fund for Distinguished Young

Scholars (41425020). This work was also partly supported by the high-performance grid-computing platform of Sun Yat-sen University. The authors thank J. Zheng of Jinan University for providing Guangdong emission inventory, and Xiang Ding of Guangzhou Institute of Geochemistry, Chinese Academy of Sciences for providing measurement data of SOA at the WQS site.



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





**Table 1: Datasets of all input parameters used in the emission inventory model**

| source | $F_{OC}$ | OM/OC | O/C | H/C | N/C | $E_{SVOCs}/E_{POA}$ | $E_{IVOCs}/E_{POA}$ |
|---|---|---|---|---|---|---|---|
| industry[m] | 0.03[a] | 1.77[d] | 0.19[d] | 1.26[d] | 0.008[d] | - | - |
| | 0.3[a] | 1.91[e] | 0.56[e] | 1.61[e] | 0.02[e] | - | - |
| | … | … | … | … | … | … | … |
| residential sources[n] | 0.1[a] | 1.39[e] | 0.17[e] | 1.8[e] | 0.004[e] | - | - |
| | 0.3[a] | 1.44[f] | 0.19[f] | 1.78[f] | 0.036[f] | - | - |
| | … | … | … | … | … | … | … |
| on-road mobile sources | 0.58[a] | 1.4[f] | 0.15[f] | 1.77[f] | 0.045[f] | 0.67[h] | 8[k] |
| | 0.37[b] | 1.46[g] | 0.19[g] | 1.78[g] | 0.05[g] | 0.49[i] | 30[j] |
| | … | … | … | … | … | … | … |
| off-road mobile sources[o] | 0.33[a] | - | - | - | - | - | - |
| | 0.32[b] | - | - | - | - | - | - |
| | … | … | … | … | … | … | … |
| dust[p] | 0.1[a] | - | - | - | - | - | - |
| | 0.05[a] | - | - | - | - | - | - |
| | … | … | … | … | … | … | … |
| biomass burning | 0.58[c] | 1.55[d] | 0.26[d] | 1.62[d] | 0.06[d] | 0.65[i] | 0.75[l] |
| | 0.6[a] | 1.62[c] | 0.32[c] | 1.47[c] | 0.06[c] | 0.8[j] | 1.5[j] |
| | … | … | … | … | … | … | … |

a. (Li et al., 2017b); b. (Zhao et al., 2011); c. (He et al., 2011); d. (Huang et al., 2011); e. (Hu et al., 2016); f. (Xu et al., 2015); g. (Ye et al., 2017); h. (May, Presto, et al., 2013a); i. (Louvaris et al., 2017); j. (Zhao et al., 2016a); k. (Zhao et al., 2015); l. (Shrivastava et al., 2008), etc. m. data of ratios of S/IVOCs to POA for industry are the same as those for on-road mobile sources; n. data of ratios of S/IVOCs to POA for residential sources are the same as those for on-road mobile sources; o. data of ratios of S/IVOCs to POA, O/C, H/C, N/C, and OM/OC for off-road mobile sources are the same as those for on-road mobile sources; p. data of ratios of S/IVOCs to POA for dust are the same as those for on-road mobile sources; data of ratios of O/C, H/C, N/C, and OM/OC for dust are the same as those for industry.



**Table 2: Probabilistic distributions with uncertainty range at the 95% confidence interval in model input parameters**

| input parameters | source | distribution type | para1 | para2 | mean value | uncertainty range (95% confidence level) |
|---|---|---|---|---|---|---|
| $F_{OC}$ | industry | weibull | 0.09 | 1.07 | 0.08 | (0.005, 0.28) |
| | residential sources | normal | 0.46 | 0.17 | 0.45 | (0.17, 0.70) |
| | on-road mobile sources | weibull | 0.39 | 2.02 | 0.33 | (0.07, 0.70) |
| | off-road mobile sources | normal | 0.26 | 0.11 | 0.25 | (0.06, 0.70) |
| | dust | weibull | 0.08 | 5.26 | 0.08 | (0.05, 0.10) |
| | biomass burning | lognormal | -1.01 | 0.34 | 0.38 | (0.19, 0.68) |
| OM/OC | industry | gamma | 111.46 | 0.02 | 1.69 | (1.43, 1.94) |
| | residential sources | lognormal | 0.28 | 0.05 | 1.33 | (1.26, 1.43) |
| | on-road mobile sources | lognormal | 0.34 | 0.05 | 1.39 | (1.31, 1.46) |
| | off-road mobile sources | lognormal | 0.34 | 0.05 | 1.39 | (1.31, 1.46) |
| | dust | gamma | 111.46 | 0.02 | 1.69 | (1.43, 1.94) |
| | biomass burning | lognormal | 0.43 | 0.09 | 1.51 | (1.40, 1.61) |
| $E_{SVOCs}/E_{POA}$ | industry | lognormal | -0.32 | 0.23 | 0.70 | (0.51, 0.97) |
| | residential sources | lognormal | -0.32 | 0.23 | 0.70 | (0.51, 0.97) |
| | on-road mobile sources | lognormal | -0.32 | 0.23 | 0.70 | (0.51, 0.97) |
| | off-road mobile sources | lognormal | -0.32 | 0.23 | 0.70 | (0.51, 0.97) |
| | dust | lognormal | -0.32 | 0.23 | 0.70 | (0.51, 0.97) |
| | biomass burning | normal | 0.76 | 0.14 | 0.80 | (0.58, 0.97) |
| $E_{IVOCs}/E_{POA}$ | industry | lognormal | 1.86 | 0.88 | 8.00 | (1.79, 25.45) |
| | residential sources | lognormal | 1.86 | 0.88 | 8.00 | (1.79, 25.45) |
| | on-road mobile sources | lognormal | 1.86 | 0.88 | 8.00 | (1.79, 25.45) |
| | off-road mobile sources | lognormal | 1.86 | 0.88 | 8.00 | (1.79, 25.45) |
| | dust | lognormal | 1.86 | 0.88 | 8.00 | (1.79, 25.45) |
| | biomass burning | gamma | 0.66 | 0.82 | 0.40 | (0.002, 1.33) |
| O/C | industry | weibull | 0.49 | 2.70 | 0.44 | (0.19, 0.73) |
| | residential sources | normal | 0.13 | 0.05 | 0.13 | (0.08, 0.19) |
| | on-road mobile sources | lognormal | -1.84 | 0.26 | 0.16 | (0.11, 0.21) |
| | off-road mobile sources | lognormal | -1.84 | 0.26 | 0.16 | (0.11, 0.21) |
| | dust | weibull | 0.49 | 2.70 | 0.44 | (0.19, 0.73) |
| | biomass burning | lognormal | -1.29 | 0.35 | 0.30 | (0.19, 0.47) |
| H/C | industry | gamma | 71.81 | 0.02 | 1.59 | (1.30, 1.90) |





| | | | para1 | para2 | | |
|---|---|---|---|---|---|---|
| | residential sources | weibull | 1.76 | 32.93 | 1.72 | (1.60, 1.80) |
| | on-road mobile sources | weibull | 1.77 | 90.92 | 1.75 | (1.71, 1.78) |
| | off-road mobile sources | weibull | 1.77 | 90.92 | 1.75 | (1.71, 1.78) |
| | dust | gamma | 71.81 | 0.02 | 1.59 | (1.30, 1.90) |
| | biomass burning | lognormal | 0.45 | 0.05 | 1.55 | (1.48, 1.62) |
| | industry | lognormal | -4.02 | 0.76 | 0.02 | (0.01, 0.07) |
| | residential sources | lognormal | -4.45 | 1.01 | 0.02 | (0.00, 0.05) |
| N/C | on-road mobile sources | normal | 0.03 | 0.02 | 0.03 | (0.01, 0.05) |
| | off-road mobile sources | normal | 0.03 | 0.02 | 0.03 | (0.01, 0.05) |
| | dust | lognormal | -4.02 | 0.76 | 0.02 | (0.01, 0.07) |
| | biomass burning | lognormal | -3.62 | 0.81 | 0.03 | (0.01, 0.06) |

para1: the mean for normal, the mean of lnx for lognormal, the scale parameter for gamma and weibull distributions.

para2: the standard deviation for normal, the standard deviation of lnx for lognormal, the shape parameter for gamma and weibull distributions.



**Table 3: WRF-Chem Model Configuration**

| | |
|---|---|
| Simulation period | 19-25 November 2008 |
| Simulated regional center | (22.7°N,113.6°E) |
| Number of grids | 148×139 |
| Horizontal resolution | 3km×3km |
| Vertical levels | 40 |
| Meteorological initial and boundary conditions | ECMWF |
| Chemical boundary conditions | MOZART |
| Microphysics | Lin |
| Longwave Radiation | RRTM |
| Shortwave Radiation | New Goddard |
| Surface Layer | Monin-Obukhov (Janjic) scheme |
| Land Surface | Noah LSM |
| Urban Surface | 3-category UCM |
| Planetary Boundary layer | Yonsei University scheme |
| Cumulus | New Grell scheme(G3) |
| Photolysis | Fast-J |
| Gas-phase chemistry | SAPRC99 |
| Aerosol chemistry | MOSAIC-4bins for inorganic aerosols; Simplified volatility basis set (VBS) for organic aerosols |
| Sea salt emissions | MOSAIC or MADE/SORGAM sea salt emissions |
| Biogenic emissions | MEGAN |





**Table 4: S/IVOCs emission inventory in the PRD region for the year 2010**

| source | S/IVOC emission (Gg/year) | contribution (%) |
|---|---|---|
| industry | 114.6 | 35.4 |
| residential sources | 8.4 | 2.6 |
| on-road mobile sources | 134.4 | 41.6 |
| off-road mobile sources | 4.8 | 1.5 |
| dust | 46.8 | 14.5 |
| biomass burning | 14.4 | 4.5 |
| total | 323.4 | 100.0 |

**Table 5: Uncertainty assessment of the S/IVOCs emission inventory for the year 2010**

| source | Correlation coefficients between model inputs and outputs (Uncertainty range at 95% confidence interval) | | | | | | | |
|---|---|---|---|---|---|---|---|---|
| | $E_{SVOCs}$ | $E_{IVOCs}$ | $E_{S/IVOCs}$ | $F_{OC}$ | OM/OC | $E_{PM2.5}$ | $E_{SVOCs}/E_{POA}$ | $E_{IVOCs}/E_{POA}$ |
| industry | 0.108 (-95%, 283%) | 0.510 (-97%, 404%) | 0.496 (-97%, 386%) | 0.098 (-94%, 233%) | 0.006 (-16%, 15%) | 0.018 (-54%, 55%) | | |
| residential | 0.060 (-78%, 132%) | 0.631 (-89%, 281%) | 0.618 (-88%, 264%) | 0.035 (-63%, 56%) | 0.005 (-6%, 7%) | 0.032 (-66%, 66%) | | |
| on-road mobile | 0.477 (-87%, 183%) | 0.961 (-93%, 321%) | **0.956** **(-92%, 302%)** | **0.345** **(-79%, 103%)** | 0.020 (-6%, 5%) | **0.204** **(-73%, 73%)** | 0.019 (-29%, 34%) | **0.782** **(-79%, 201%)** |
| off-road mobile | 0.023 (-83%, 137%) | 0.589 (-91%, 284%) | 0.575 (-90%, 266%) | 0.012 (-78%, 70%) | 0.001 (-6%, 5%) | 0.011 (-59%, 59%) | | |
| dust | 0.078 (-68%, 103%) | 0.692 (-86%, 252%) | 0.682 (-84%, 235%) | 0.033 (-35%, 28%) | 0.015 (-16%, 15%) | 0.041 (-61%, 61%) | | |
| biomass burning | 0.027 (-70%, 127%) | 0.026 (-100%, 336%) | 0.032 (-75%, 163%) | 0.016 (-50%, 79%) | 0.001 (-7%, 7%) | 0.092 (-58%, 58%) | 0.012 (-25%, 26%) | 0.013 (-99%, 247%) |
| total | (-55%, 90%) | **(-85%, 250%)** | **(-79%, 229%)** | | | | | |



**Table 6: Overview of simulations**

| Test Name | S/IVOCs emission inventory | $k_{OH}$ of S/IVOCs (cm$^3$ molecule$^{-1}$ s$^{-1}$) | Notes |
|---|---|---|---|
| BASE | No S/IVOC emissions | | |
| CASE1 | All anthropogenic S/IVOC emissions | $0.57 \times 10^{-11}$ | To evaluate the effect of $k_{OH}$ on the formation of SOA. |
| CASE2 | All anthropogenic S/IVOC emissions | $0.42 \times 10^{-11}$ | |
| CASE3 | 21% of the S/IVOC emissions in CASE2 | $0.42 \times 10^{-11}$ | To evaluate the sensitivity of SOA simulation to S/IVOCs emission. |
| CASE4 | 45% of the S/IVOC emissions in CASE2 | $0.42 \times 10^{-11}$ | |
| CASE5 | 1.27 times the S/IVOC emissions in CASE2 | $0.42 \times 10^{-11}$ | |
| CASE6 | 3.29 times the S/IVOC emissions in CASE2 | $0.42 \times 10^{-11}$ | |
| CASE7 | Only S/IVOC emissions from biomass burning | $0.42 \times 10^{-11}$ | To quantify the contributions of S/IVOCs emitted from different source categories to the formation of SOA. |
| CASE8 | Only S/IVOC emissions from dust | $0.42 \times 10^{-11}$ | |
| CASE9 | Only industrial S/IVOC emissions | $0.42 \times 10^{-11}$ | |
| CASE10 | Only off-road mobile S/IVOC emissions | $0.42 \times 10^{-11}$ | |
| CASE11 | Only on-road mobile S/IVOC emissions | $0.42 \times 10^{-11}$ | |
| CASE12 | Only residential S/IVOC emissions | $0.42 \times 10^{-11}$ | |



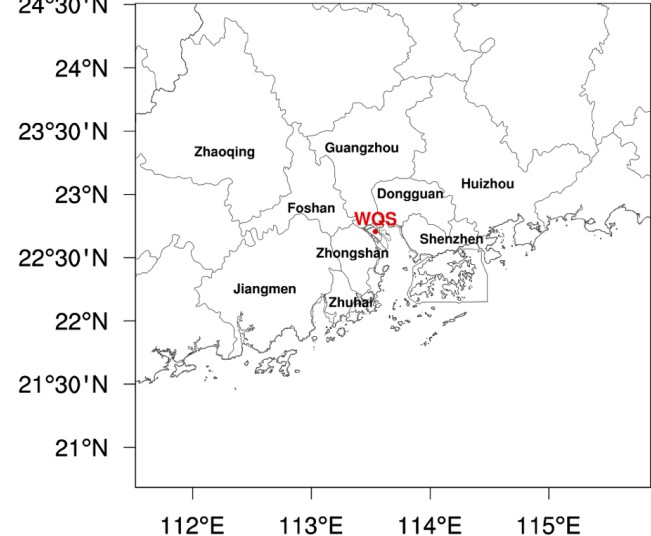

**Figure 1: Modeling domain and locations of Wan Qing Sha air quality monitoring sites.**

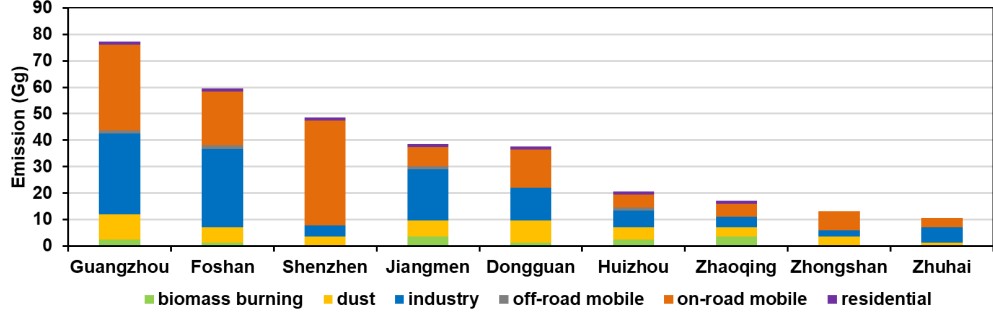

5                                    **Figure 2: Source specific emissions in each city for the year 2010.**



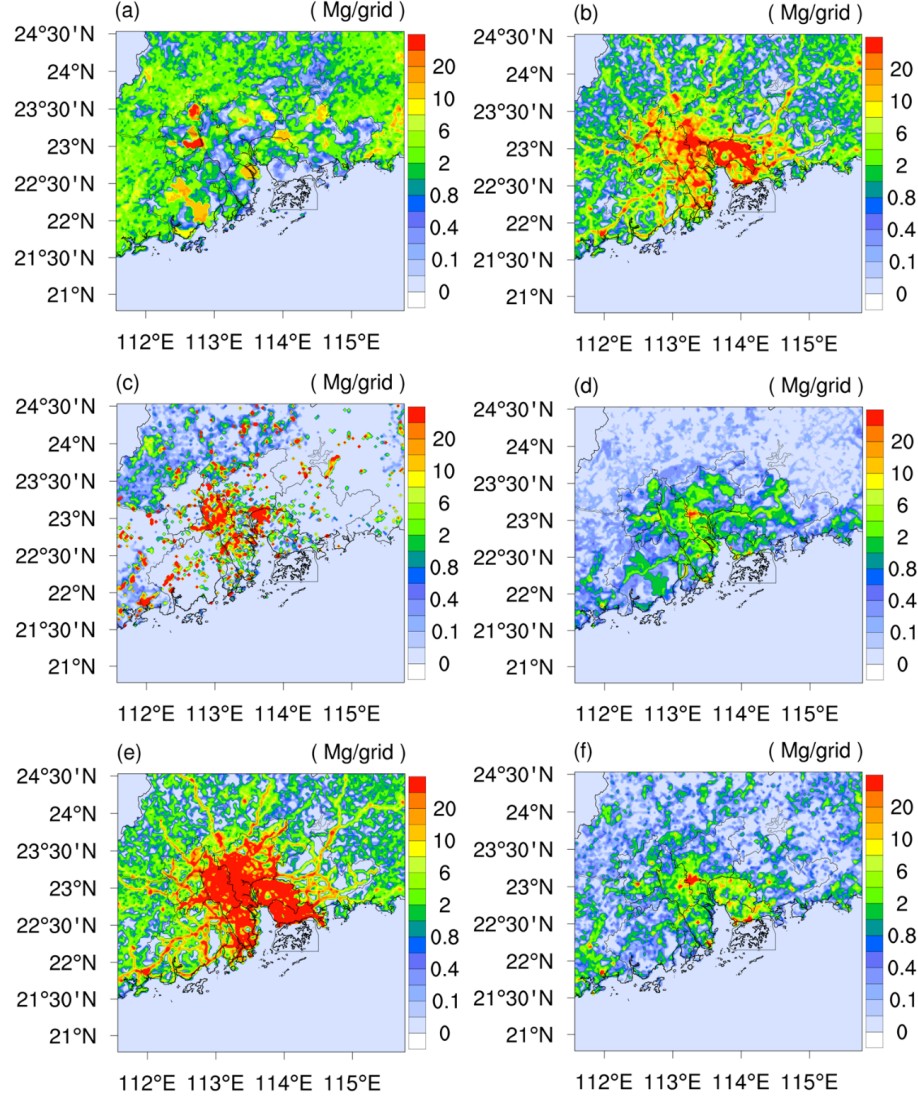

**Figure 3: Spatial distribution of S/IVOC emissions from different source categories for the year 2010: (a) biomass burning (b) dust (c) industry (d) off-road mobile sources (e) on-road mobile sources (f) residential sources.**




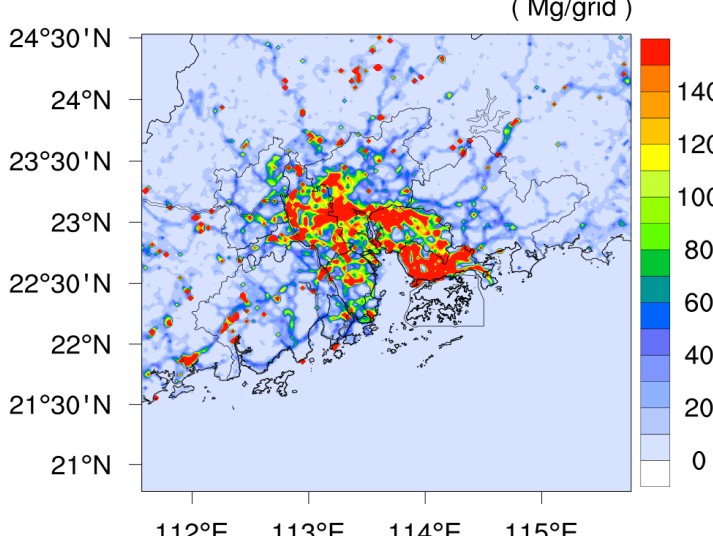

**Figure 4: Spatial distribution of total S/IVOC emissions for the year 2010.**

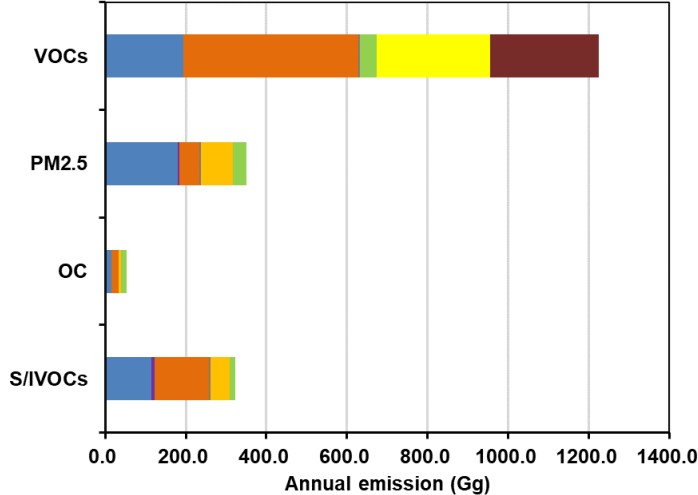

**Figure 5: Comparisons with emissions of other pollutants in the PRD region for the year 2010.**

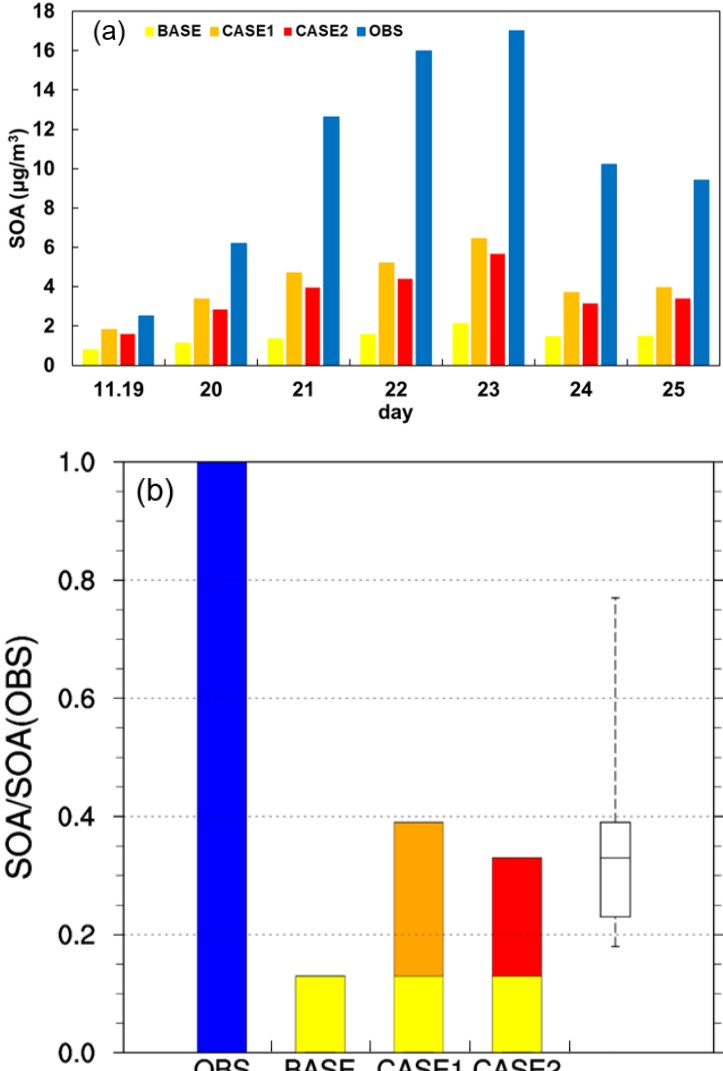

**Figure 6: Comparisons of SOA between simulations and observations at WQS monitoring site: (a) time series; (b) ratios of**
5   **temporal average SOA concentration to observed SOA concentration during the study period (the box represents the uncertainty**
**range in SOA prediction, the central line is the ratio in CASE2, the edges of the box are the ratios in CASE4 and CASE5, the edges**
**of the whisker are the ratios in CASE3 and CASE6).**


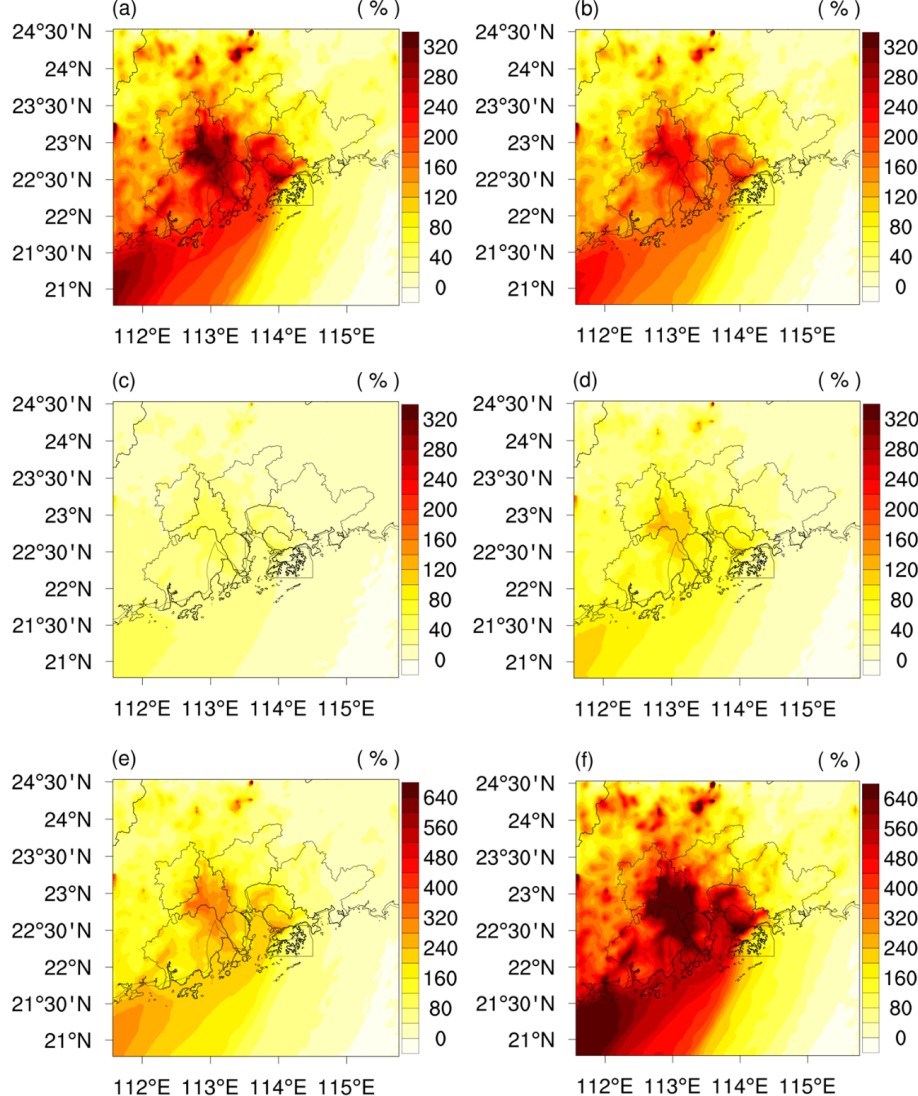

**Figure 7: Relative difference of SOA between CASE runs and BASE: (a) CASE1, (b) CASE2, (c) CASE3, (d) CASE4, (e) CASE5, (f) CASE6.**

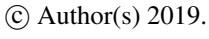



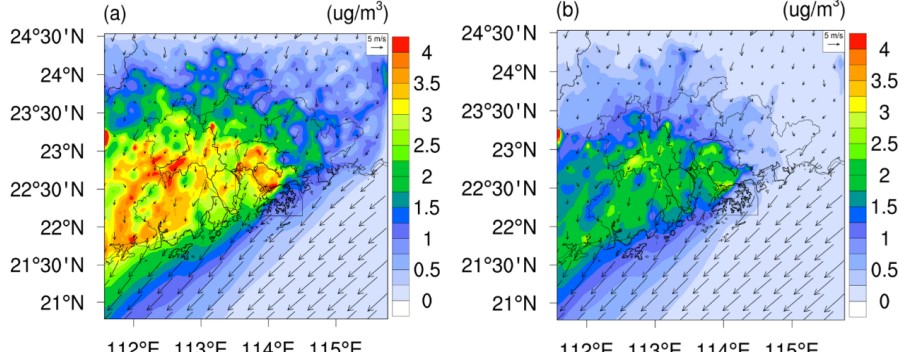

**Figure 8: Spatial distribution of temporal average (a) SOA and (b) SI-SOA during the study period over the modeling domain in CASE2 run.**



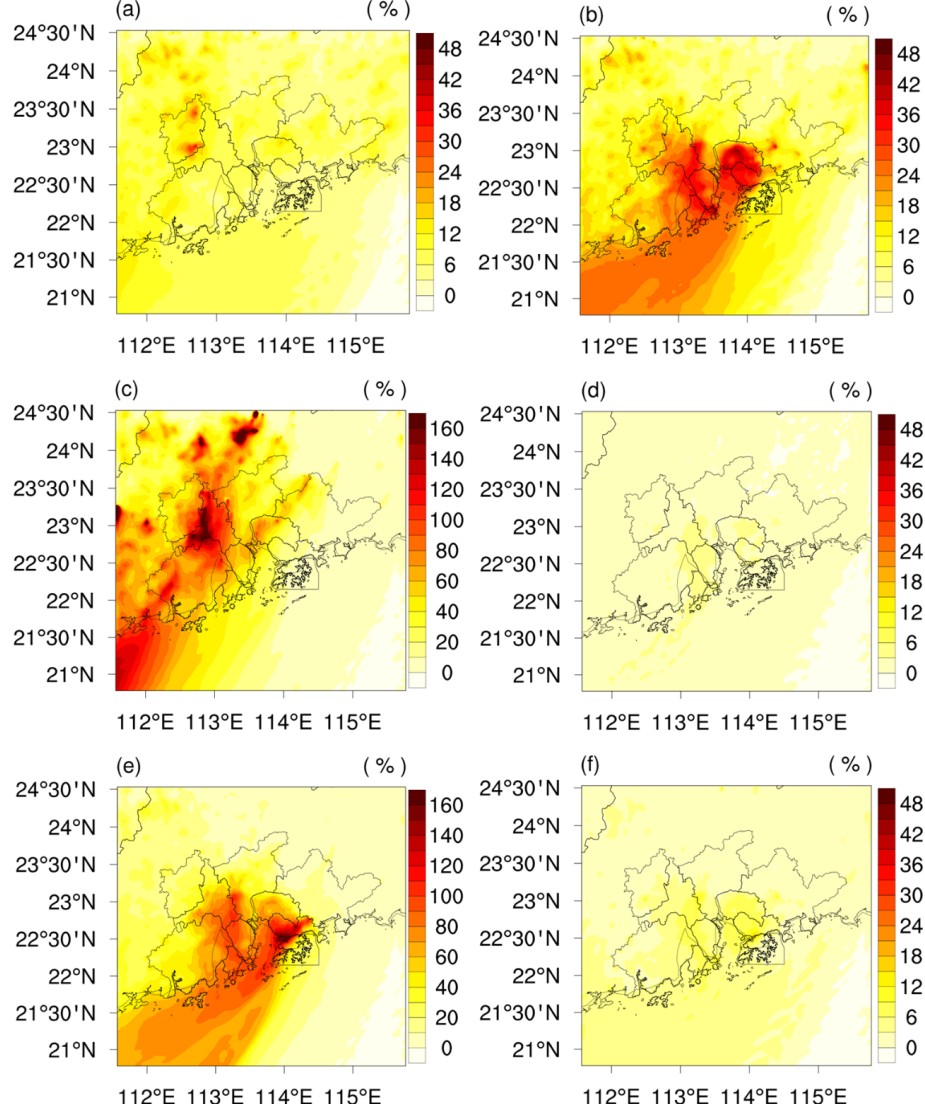

**Figure 9: Relative difference of SOA between CASE runs and BASE: (a) CASE7, (b) CASE8, (c) CASE9, (d) CASE10, (e) CASE11, (f) CASE12.**