# Peer review of "Emission inventory of semi-volatile and intermediate volatility organic compounds and their effects on SOA over the Pearl River Delta region"

_Atmospheric Chemistry and Physics, 2018_

## Referee Comment (RC1) · Anonymous Referee #2 · 12 Apr 2019

Comments on "Emission inventory of semi-volatile and intermediate volatility organic compounds and their effects on SOA over the Pearl River Delta region"

**General comments:**

The manuscript by Wu et al. gave an overview of S/IVOCs and their contributions to SOA formation based on the model simulation. The authors improved the model setup parameters and reduced the uncertainty of simulation. The improved model simulation was used to evaluate the effect of S/IVOCs and key anthropogenic S/IVOCs to SOA formation. And the results also showed the potential area of S/IVOCs and sources.

Overall, the manuscript is well organized and within the scope of Atmospheric Chemistry and Physics. I have some minor comments about the manuscript. After addressing the concerns, I would recommend this manuscript for publication.

**Specific comments:**

1. Page 5 line 19. What are the POA factors and OM/OC ratios of different sources? Could you please provide the detailed information about them?

2. Page 8 line 15. Please clarify the definition of SI-SOA.

3. Page 9 line 6. I wonder where the OH rate constants come from. Please explain it.

4. Page 12 line 4. From Fig. 2, I find that the dust and industry contributions to Zhaoqing and Shenzhen are similar. Do you consider the uncertainty when estimate the emission inventories?

5. Page 15 line 9. "The simulation results of SOA formation" will be better.

6. Have you tried to screen out the dominant species among S/IVOCs which contribute mostly to SOA formation in PRD region?

**Technical corrections:**

1. Page 1 line 15. "emissions" may be "emission".

2. Page 2 line 12. "secondary organic aerosols" should be "secondary organic aerosol".

3. Page 2 line 13. Please add some refs. About the SOA contribution to PM2.5.

4. Page 2 line 22. The ref. "Guo et al. et al.," should be "Guo et al.,". Please revise it.

5. Page 12 line 7. I think you have miswritten the figure number. Fig. 4 may be Fig. 3? Please check it.

6. Page 12 line 14. The discussion order is weird. Fig.3a is after Fig. 3c. And Fig.3b is after Fig. 3d. Please reorganize the discussion about spatial distribution of S/IVOCs.

7. Page 14. In my opinion, I think the whole paragraph on this page is discussing the S/IVOCs emission inventory and comparisons with another study. So maybe an addition of a section title (3.3 …) here will be better.

---

## Referee Comment (RC2) · Anonymous Referee #1 · 17 Apr 2019

The authors developed a high resolution emission inventory of semi-volatile and intermediate volatility organic compounds (S/IVOCs) for the Pearl River Delta region, and then evaluated the impacts of anthropogenic S/IVOCs on secondary organic aerosols (SOA) by a regional chemical transport model. The primary emissions and chemical degradation mechanisms of S/IVOCs are among the key knowledge gaps in better understanding and predicting the SOA formation. Thus this study is useful for future modelling studies of SOA in the PRD region. I would recommend that this manuscript can be considered for publication after the following specific comments being addressed.

Introduction: a brief introduction of S/IVOCs at its first appearance should be helpful

Discussion paper

for the readership to better understand the context of this study. For example, what compounds do the S/IVOCs include? What are their major sources? Etc.

Section 2.1: it is not clear if this S/IVOCs emission inventory only include one 'bulk species' or include individually many S/IVOC compounds for model use? I also wonder if there are biogenic sources of S/IVOCs. If so, the authors may need elaborate that this study mainly focused on anthropogenic emissions.

Section 3.1: I am curious that dust is a source of S/IVOCs. It would be helpful if the authors elaborate more about this source by several sentences.

P1, L13: change "the Pearl River Delta (PRD)" to "the PRD", as you have already defined the PRD in Line 12.

P1, L15: emission factors of POA...

P5, L16-17: I suggest the authors to provide a brief description of the definition of parameters used in this Equation (1) here, so that the readers do not need to refer to the supplement.

P9, L13-14: the same as the above comment.

P10, L6: the roles of S/IVOCs in the formation of SOA...

P15, L13: in my opinion, WQS should be a regional receptor site in the PRD region other than a regional background site, as it is generally located at the downwind of city clusters in winter monsoon season.

Table 3: I suggest moving this table to the supplement.

Is this newly developed emission inventory can be used by the modelling community? If so, how can it be accessed? A comment on this should be given in the data availability.

---

## Author Comment (AC1) · 14 May 2019

Response to the Reviewer We appreciate the reviewer for his/her constructive criticisms and valuable comments, which were of great help in improving the quality of the manuscript. We have revised the manuscript accordingly and our detailed responses are shown below. All the revision is highlighted in the revised manuscript.

Referee 2 Comments General comments: The manuscript by Wu et al. gave an overview of S/IVOCs and their contributions to SOA formation based on the model simulation. The authors improved the model setup parameters and reduced the uncertainty of simulation. The improved model simulation was used to evaluate the effect of S/IVOCs and key anthropogenic S/IVOCs to SOA formation. And the results also showed the potential area of S/IVOCs and sources. Overall, the manuscript is well organized and within the scope of Atmospheric Chemistry and Physics. I have some minor comments about the manuscript. After addressing the concerns, I would recommend this manuscript for publication. Reply: Many thanks for the reviewer's encouraging words. We have addressed all of the comments/suggestions in the revised manuscript. Detailed responses to the individual specific comment/suggestion are as follows.

Specific comments: R2.1. Page 5 line 19. What are the POA factors and OM/OC ratios of different sources? Could you please provide the detailed information about them? Reply: Sorry for the unclear expressions. It meant that POA emission factors and OM/OC ratios for different source categories, which included the industry, on-road and off-road mobile sources, residential sources, dust and biomass burning. To provide a clearer description, the text has been revised as followed: "[. . .] POA emission factors for different source categories (e.g., industry, on-road and off-road mobile sources, residential sources, dust, and biomass burning) were obtained from POC (primary organic carbon) emission factors using source-specific ratios of OM/OC (mass ratios of organic matter to organic carbon) [. . .]" For details, please refer to Lines 11-14, Page 6 in the revised manuscript.

R2.2. Page 8 line 15. Please clarify the definition of SI-SOA. Reply: Thanks for pointing this out. The definitions of V-SOA and SI-SOA have been provided as followed: "V-SOA (SOA formed by the oxidation of VOCs-traditional SOA precursors emitted from varied anthropogenic and biogenic sources) and SI-SOA (SOA formed by the oxidation of S/IVOCs-untraditional SOA precursors emitted from anthropogenic sources)." For details, please refer to Lines 16-19, Page 9 in the revised manuscript.

R2.3. Page 9 line 6. I wonder where the OH rate constants come from. Please explain it. Reply: The reviewer's comment is highly appreciated. The OH reaction rate constant (kOH) of $0.57 \times 10^{-11}$ cm3 molecule-1 s-1 in the 2-species VBS was reduced

by a factor of 7 from that of the 9-species VBS ($4 \times 10$-11 cm3 molecule-1 s-1), which was assumed to be ~50% higher than that of a typical large saturated n-alkane as suggested by previous studies (Atkinson and Arey, 2003; Robinson et al., 2007), in order to align the SOA predictions between 2-species and 9-species VBS schemes. The above description has been provided in the revised manuscript (Page 10, Lines 10-15).

R2.4. Page 12 line 4. From Fig. 2, I find that the dust and industry contributions to Zhaoqing and Shenzhen are similar. Do you consider the uncertainty when estimate the emission inventories? Reply: Thanks for the reviewer's comment. The magnitudes of S/IVOC emissions from dust and industry in Zhaoqing and Shenzhen were similar, but the contributions were different. Dust contributed about 21.1% (3.6 Gg) and 7.4% (3.6 Gg) to the S/IVOC emissions in Zhaoqing and Shenzhen, and industry contributed about 22.8% (3.9 Gg) and 8.8% (4.3 Gg), respectively. The contributions of dust and industry to S/IVOC emissions in Shenzhen were smaller than those in Zhaoqing, attributable to the dominance of on-road mobile S/IVOC emissions in Shenzhen (81.3%, 39.6 Gg) because of the dense traffic (Pan et al., 2015). As Shenzhen and Zhaoqing have much less industrial point sources than cities located in the southeastern PRD such as southern Guangzhou and Foshan (Pan et al., 2015), their corresponding industrial S/IVOC emissions were also less. There were relatively higher S/IVOC emissions from road fugitive dust and lower emissions from building construction dust in Zhaoqing than those in Shenzhen because of shorter road lengths and more developed construction industries in Shenzhen (GSY, 2011; Peng et al., 2013), resulting in similar magnitudes of S/IVOC emissions from dust in these two cities. The above description for similar magnitudes and different contributions of S/IVOC emissions from dust and industry in Zhaoqing and Shenzhen has been provided in the revised manuscript. For details, please refer to Lines 12-24, Page 13 in the revised manuscript.

On the other hand, we indeed have considered the uncertainty when estimate the emission inventory of S/IVOCs. As Table 4 in the revised manuscript showed, the uncertainties in S/IVOC emissions from dust and industry ranged from -84% to 235% and from -97% to 386% at 95% confidence interval, respectively. And the ratios of EIVOCs/EPOA used in calculating the S/IVOC emissions for these two source categories were the key sources of uncertainties in the emission estimates. For details of the uncertainty analysis, please refer to Section 3.2 and Table 4 in the revised manuscript.

R2.5. Page 15 line 9. "The simulation results of SOA formation" will be better. Reply: Thanks for the suggestion. We have revised manuscript accordingly (Line 4, Page 17).

R2.6. Have you tried to screen out the dominant species among S/IVOCs which contribute mostly to SOA formation in PRD region? Reply: Thanks for the reviewer's great comment. As the fact that S/IVOCs in the model was treated as a 'bulk species' rather than the individual species, it is unable to figure out the dominant species among S/IVOCs that contribute mostly to SOA formation in PRD region. To better understand the roles of individual S/IVOCs in SOA formation, future work by using different models and source apportionment results would be conducted.

Technical corrections: R2.1. Page 1 line 15. "emissions" may be "emission". Reply: Sorry for the mistake. It has been revised accordingly (Line 14, Page 1).

R2.2. Page 2 line 12. "secondary organic aerosols" should be "secondary organic aerosol". Reply: Sorry for the mistake. It has been revised accordingly (Line 12, Page 2).

R2.3. Page 2 line 13. Please add some refs. About the SOA contribution to PM2.5. Reply: Sorry for the mistake. References have been added and the description of SOA as the key component of PM2.5 has been revised in the revised manuscript as followed: "As the key component, secondary organic aerosol (SOA) accounts for 20–80% of organic aerosol (OA), while OA accounts for 20%–90% of fine particulate matter (PM2.5) (Kanakidou et al., 2005; Carlton, Wiedinmyer and Kroll, 2009; Zhang et al., 2007, 2013)." For details, please refer to Lines 12-14, Page 2 in the revised manuscript.

R2.4. Page 2 line 22. The ref. "Guo et al. et al.," should be "Guo et al.,". Please revise it. Reply: Sorry for the mistake. It has been revised accordingly (Line 24, Page 2).

R2.5. Page 12 line 7. I think you have miswritten the figure number. Fig. 4 may be Fig. 3? Please check it. Reply: Sorry for the mistake. The figure number has been revised through the manuscript (Lines 26-27, Page 13).

R2.6. Page 12 line 14. The discussion order is weird. Fig.3a is after Fig. 3c. And Fig.3b is after Fig. 3d. Please reorganize the discussion about spatial distribution of S/IVOCs. Reply: Thanks for the suggestion. We have reorganized the discussion about the spatial distribution of S/IVOCs in the revised manuscript accordingly. For details, please refer to Lines 2-15, Page 14 in the revised manuscript.

R2.7. Page 14. In my opinion, I think the whole paragraph on this page is discussing the S/IVOCs emission inventory and comparisons with another study. So maybe an addition of a section title (3.3 . . .) here will be better. Reply: Thanks for the constructive suggestion. We have added a section title "3.3 Comparison with other emission inventory" in Line 23, Page 15.

Please also note the supplement to this comment:
https://www.atmos-chem-phys-discuss.net/acp-2018-1341/acp-2018-1341-AC1-supplement.pdf
* * *
[Figure]

**Supplement:**

Response to the Reviewer
We appreciate the reviewer for his/her constructive criticisms and valuable comments, which were of great help in improving the quality of the manuscript. We have revised the manuscript accordingly and our detailed responses are shown below. All the revision is highlighted in the revised manuscript.

**Referee 2 Comments**

**General comments:**

The manuscript by Wu et al. gave an overview of S/IVOCs and their contributions to SOA formation based on the model simulation. The authors improved the model setup parameters and reduced the uncertainty of simulation. The improved model simulation was used to evaluate the effect of S/IVOCs and key anthropogenic S/IVOCs to SOA formation. And the results also showed the potential area of S/IVOCs and sources.

Overall, the manuscript is well organized and within the scope of Atmospheric Chemistry and Physics. I have some minor comments about the manuscript. After addressing the concerns, I would recommend this manuscript for publication.

Reply: Many thanks for the reviewer's encouraging words. We have addressed all of the comments/suggestions in the revised manuscript. Detailed responses to the individual specific comment/suggestion are as follows.

**Specific comments:**

**R2.1.** Page 5 line 19. What are the POA factors and OM/OC ratios of different sources? Could you please provide the detailed information about them?

Reply: Sorry for the unclear expressions. It meant that POA emission factors and OM/OC ratios for different source categories, which included the industry, on-road and off-road mobile sources, residential sources, dust and biomass burning. To provide a clearer description, the text has been revised as followed:

"[…] POA emission factors for different source categories (*e.g.*, industry, on-road and off-road mobile sources, residential sources, dust, and biomass burning) were

obtained from POC (primary organic carbon) emission factors using source-specific ratios of OM/OC (mass ratios of organic matter to organic carbon) […]"

For details, please refer to Lines 11-14, Page 6 in the revised manuscript.

**R2.2.** Page 8 line 15. Please clarify the definition of SI-SOA.

Reply: Thanks for pointing this out. The definitions of V-SOA and SI-SOA have been provided as followed:

"V-SOA (SOA formed by the oxidation of VOCs-traditional SOA precursors emitted from varied anthropogenic and biogenic sources) and SI-SOA (SOA formed by the oxidation of S/IVOCs-untraditional SOA precursors emitted from anthropogenic sources)."

For details, please refer to Lines 16-19, Page 9 in the revised manuscript.

**R2.3.** Page 9 line 6. I wonder where the OH rate constants come from. Please explain it.

Reply: The reviewer's comment is highly appreciated. The OH reaction rate constant ($k_{OH}$) of $0.57 \times 10^{-11}$ cm$^3$ molecule$^{-1}$ s$^{-1}$ in the 2-species VBS was reduced by a factor of 7 from that of the 9-species VBS ($4 \times 10^{-11}$ cm$^3$ molecule$^{-1}$ s$^{-1}$), which was assumed to be ~50% higher than that of a typical large saturated $n$-alkane as suggested by previous studies (Atkinson and Arey, 2003; Robinson et al., 2007), in order to align the SOA predictions between 2-species and 9-species VBS schemes.

The above description has been provided in the revised manuscript (Page 10, Lines 10-15).

**R2.4.** Page 12 line 4. From Fig. 2, I find that the dust and industry contributions to Zhaoqing and Shenzhen are similar. Do you consider the uncertainty when estimate the emission inventories?

Reply: Thanks for the reviewer's comment. The magnitudes of S/IVOC emissions from dust and industry in Zhaoqing and Shenzhen were similar, but the contributions were different. Dust contributed about 21.1% (3.6 Gg) and 7.4% (3.6 Gg) to the S/IVOC emissions in Zhaoqing and Shenzhen, and industry contributed about 22.8% (3.9 Gg) and 8.8% (4.3 Gg), respectively. The contributions of dust and industry to S/IVOC emissions in Shenzhen were smaller than those in Zhaoqing, attributable to the dominance of on-road mobile S/IVOC emissions in Shenzhen (81.3%, 39.6 Gg) because of the dense traffic (Pan et al., 2015). As Shenzhen and Zhaoqing have much less industrial point sources than cities located in the southeastern PRD such as southern Guangzhou and Foshan (Pan et al., 2015), their corresponding industrial S/IVOC emissions were also less. There were relatively higher S/IVOC emissions from road fugitive dust and lower emissions from building construction dust in Zhaoqing than those in Shenzhen because of shorter road lengths and more developed construction industries in Shenzhen (GSY, 2011; Peng et al., 2013), resulting in similar magnitudes of S/IVOC emissions from dust in these two cities. The above description for similar magnitudes and different contributions of S/IVOC emissions from dust and industry in Zhaoqing and Shenzhen has been provided in the revised manuscript.

For details, please refer to Lines 12-24, Page 13 in the revised manuscript.

On the other hand, we indeed have considered the uncertainty when estimate the emission inventory of S/IVOCs. As Table 4 in the revised manuscript showed, the uncertainties in S/IVOC emissions from dust and industry ranged from -84% to 235% and from -97% to 386% at 95% confidence interval, respectively. And the ratios of $E_{IVOCs}/E_{POA}$ used in calculating the S/IVOC emissions for these two source categories were the key sources of uncertainties in the emission estimates.

For details of the uncertainty analysis, please refer to Section 3.2 and Table 4 in the revised manuscript.

**R2.5.** Page 15 line 9. "The simulation results of SOA formation" will be better.

Reply: Thanks for the suggestion. We have revised manuscript accordingly (Line 4, Page 17).

**R2.6.** Have you tried to screen out the dominant species among S/IVOCs which contribute mostly to SOA formation in PRD region?

Reply: Thanks for the reviewer's great comment. As the fact that S/IVOCs in the model was treated as a 'bulk species' rather than the individual species, it is unable to figure out the dominant species among S/IVOCs that contribute mostly to SOA formation in PRD region. To better understand the roles of individual S/IVOCs in SOA formation, future work by using different models and source apportionment

results would be conducted.

**Technical corrections:**

**R2.1.** Page 1 line 15. "emissions" may be "emission".

Reply: Sorry for the mistake. It has been revised accordingly (Line 14, Page 1).

**R2.2.** Page 2 line 12. "secondary organic aerosols" should be "secondary organic aerosol".

Reply: Sorry for the mistake. It has been revised accordingly (Line 12, Page 2).

**R2.3.** Page 2 line 13. Please add some refs. About the SOA contribution to PM2.5.

Reply: Sorry for the mistake. References have been added and the description of SOA as the key component of $PM_{2.5}$ has been revised in the revised manuscript as followed:

"As the key component, secondary organic aerosol (SOA) accounts for 20–80% of organic aerosol (OA), while OA accounts for 20%–90% of fine particulate matter ($PM_{2.5}$) (Kanakidou et al., 2005; Carlton, Wiedinmyer and Kroll, 2009; Zhang et al., 2007, 2013)."

For details, please refer to Lines 12-14, Page 2 in the revised manuscript.

**R2.4.** Page 2 line 22. The ref. "Guo et al. et al.," should be "Guo et al.,". Please revise it.

Reply: Sorry for the mistake. It has been revised accordingly (Line 24, Page 2).

**R2.5.** Page 12 line 7. I think you have miswritten the figure number. Fig. 4 may be Fig. 3? Please check it.

Reply: Sorry for the mistake. The figure number has been revised through the manuscript (Lines 26-27, Page 13).

**R2.6.** Page 12 line 14. The discussion order is weird. Fig.3a is after Fig. 3c. And Fig.3b is after Fig. 3d. Please reorganize the discussion about spatial distribution of S/IVOCs.

Reply: Thanks for the suggestion. We have reorganized the discussion about the spatial distribution of S/IVOCs in the revised manuscript accordingly. For details, please refer to Lines 2-15, Page 14 in the revised manuscript.

**R2.7.** Page 14. In my opinion, I think the whole paragraph on this page is discussing the S/IVOCs emission inventory and comparisons with another study. So maybe an addition of a section title (3.3 …) here will be better.

Reply: Thanks for the constructive suggestion. We have added a section title "3.3 Comparison with other emission inventory" in Line 23, Page 15.

Reference

Atkinson, R. and Arey, J.: Gas-phase tropospheric chemistry of biogenic volatile organic compounds: a review, Atmos. Environ., 37, S197–S219, doi:10.1016/s1352-2310(03)00391-1, 2003.

Bureau of Statistics of Guangdong: Guangdong Statistical Yearbook 2011 (GSY).

China Statistics Press, Beijing, 2011 (in Chinese).

Carlton, A. G., Wiedinmyer, C. and Kroll, J. H.: A review of Secondary organic aerosol (SOA) formation from isoprene, Atmos. Chem. Phys., 9(14), 4987–5005, doi:10.5194/acp-9-4987-2009, 2009.

Chen, J., Wang, W., Liu, H. and Ren, L.: Determination of road dust loadings and chemical characteristics using resuspension, Environ. Monit. Assess., 184(3), 1693–1709, doi:10.1007/s10661-011-2071-1, 2012.

Czech, H., Sippula, O., Kortelainen, M., Tissari, J., Radischat, C., Passig, J., Streibel, T., Jokiniemi, J. and Zimmermann, R.: On-line analysis of organic emissions from residential wood combustion with single-photon ionisation time-of-flight mass spectrometry (SPI-TOFMS), Fuel, 177, 334–342, doi: 10.1016/j.fuel.2016.03.036, 2016.

Ding, X., Wang, X.-M., Gao, B., Fu, X.-X., He, Q.-F., Zhao, X.-Y., Yu, J.-Z. and Zheng, M.: Tracer-based estimation of secondary organic carbon in the Pearl River Delta, south China, J. Geophys. Res., 117(D5), D05313, doi:10.1029/2011JD016596, 2012.

Dong, T. T. T. and Lee, B. K.: Characteristics, toxicity, and source apportionment of polycylic aromatic hydrocarbons (PAHs) in road dust of Ulsan, Korea, Chemosphere, 74(9), 1245–1253, doi:10.1016/j.chemosphere.2008.11.035, 2009.

Drozd, G. T., Zhao, Y., Saliba, G., Frodin, B., Maddox, C., Oliver Chang, M. C., Maldonado, H., Sardar, S., Weber, R. J., Robinson, A. L., and Goldstein, A. H.: Detailed Speciation of Intermediate Volatility and Semivolatile Organic Compound Emissions from Gasoline Vehicles: Effects of Cold-Starts and Implications for Secondary Organic Aerosol Formation, Environ. Sci. Technol., 53(3), 1706–1714, doi:10.1021/acs.est.8b05600, 2019.

Guo, H., Jiang, F., Cheng, H. R., Simpson, I. J., Wang, X. M., Ding, A. J., Wang, T. J., Saunders, S. M., Wang, T., Lam, S. H. M., Blake, D. R., Zhang, Y. L. and Xie, M.: Concurrent observations of air pollutants at two sites in the Pearl River Delta and the implication of regional transport, Atmos.Chem. Phys., 9, 7343–7360, doi:10.5194/acp-9-7343-2009, 2009.

Hodzic, A., Jimenez, J. L., Madronich, S., Canagaratna, M. R., Decarlo, P. F., Kleinman, L. and Fast, J.: Modeling organic aerosols in a megacity: Potential contribution of semi-volatile and intermediate volatility primary organic compounds to secondary organic aerosol formation, Atmos. Chem. Phys., 10, 5491–5514, doi:10.5194/acp-10-5491-2010, 2010.

Jathar, S. H., Miracolo, M. A., Presto, A. A., Donahue, N. M., Adams, P. J. and Robinson, A. L.: Modeling the formation and properties of traditional and non-traditional secondary organic aerosol: Problem formulation and application to aircraft exhaust, Atmos. Chem. Phys., 12(19), 9025–9040, doi:10.5194/acp-12-9025-2012, 2012.

Kanakidou, M., Seinfeld, J. H., Pandis, S. N., Barnes, I., Dentener, F. J., Facchini, M. C., Van Dingenen, R., et al.: Organic aerosol and global climate modelling: a review, Atmos. Chem. Phys., 5(4), 1053–1123, doi:10.5194/acp-5-1053-2005, 2005.

Khare, P. and Gentner, D. R.: Considering the future of anthropogenic gas-phase organic compound emissions and the increasing influence of non-combustion sources on urban air quality, Atmos. Chem. Phys. 18, 5391–5413, doi:10.5194/acp-18-5391-2018, 2018.

Koo, B., Knipping, E., Yarwood, G.: 1.5-Dimensional volatility basis set approach for modeling organic aerosol in CAMx and CMAQ, Atmos. Environ., 95, 158–164, doi:10.1016/j.atmosenv.2014.06.031, 2014.

Li, W., Li, L., Chen, C. li, Kacarab, M., Peng, W., Price, D., Xu, J. and Cocker, D. R.: Potential of select intermediate-volatility organic compounds and consumer products for secondary organic aerosol and ozone formation under relevant urban conditions, Atmos. Environ., 178, 109–117, doi:10.1016/j.atmosenv.2017.12.019, 2018.

May, A. A., Presto, A. A., Hennigan, C. J., Nguyen, N. T., Gordon, T. D. and Robinson, A. L.: Gas-particle partitioning of primary organic aerosol emissions: (1) Gasoline vehicle exhaust, Atmos. Environ., 77, 128–139, doi:10.1016/j.atmosenv.2013.04.060, 2013a.

May, A. A., Presto, A. A., Hennigan, C. J., Nguyen, N. T., Gordon, T. D. and Robinson, A. L.: Gas-particle partitioning of primary organic aerosol emissions: (2) diesel vehicles, Environ. Sci. Technol., 47(15), 8288–8296, doi:10.1021/es400782j, 2013b.

Ots, R., Young, D. E., Vieno, M., Xu, L., Dunmore, R. E., Allan, J. D., Coe, H., Williams, L. R., Herndon, S. C., Ng, N. L., Hamilton, J. F., Bergström, R., Di Marco, C., Nemitz, E., Mackenzie, I. A., Kuenen, J. J. P., Green, D. C., Reis, S., and Heal, M. R.: Simulating secondary organic aerosol from missing diesel-related intermediate-volatility organic compound emissions during the Clean Air for London (ClearfLo) campaign, Atmos. Chem. Phys., 16, 6453-6473, doi:10.5194/acp-16-6453-2016, 2016.

Palm, B. B., Campuzano-Jost, P., Ortega, A. M., Day, D. A., Kaser, L., Jud, W., Karl, T., Hansel, A., Hunter, J. F., Cross, E. S., Kroll, J. H., Peng, Z., Brune, W. H., and Jimenez, J. L.: In situ secondary organic aerosol formation from ambient pine forest air using an oxidation flow reactor, Atmos. Chem. Phys., 16, 2943-2970, doi:10.5194/acp-16-2943-2016, 2016.

Palm, B. B., Campuzano-Jost, P., Day, D. A., Ortega, A. M., Fry, J. L., Brown, S. S., Zarzana, K. J., Dube, W., Wagner, N. L., Draper, D. C., Kaser, L., Jud, W., Karl, T., Hansel, A., Gutiérrez-Montes, C., and Jimenez, J. L.: Secondary organic aerosol formation from in situ OH, O3, and NO3 oxidation of ambient forest air in an oxidation flow reactor, Atmos. Chem. Phys., 17, 5331-5354, doi:10.5194/acp-17-5331-2017, 2017.

Pan, Y., Nan, L. I., Zheng, J., Yin, S., Cheng, L. I., Jing, Y., Zhong, L., Chen, D., Deng, S. and Wang, S.: Emission inventory and characteristics of anthropogenic air pollutant sources in Guangdong Province, Acta Sci. Circumstantiae, 59(1), 133–135, doi: 10.13671/j.hjkxxb.2014.1058, 2015 (in Chinese).

Peng, K., Yang, Y., Zheng, J. Y., Yin, S. S., Gao, Z. J., Huang, X. B.:(2013). Emission factor and inventory of paved road fugitive dust sources in the Pearl River Delta

region, Acta Sci. Circumstantiae, 33(10), 2657-2663, doi: 10.13671/j.hjkxxb.2013.10.011, 2013 (in Chinese).

Robinson, A. L., Donahue, N. M., Shrivastava, M. K., Weitkamp, E. A., Sage, A. M., Grieshop, A. P., Lane, T. E., Pierce, J. R. and Pandis, S. N.: Rethinking organic aerosols: Semivolatile emissions and photochemical aging, Science, 315(5816), 1259–1262, doi:10.1126/science.1133061, 2007.

Rogge, W. F., Hildemann, L. M., Mazurek, M. A., Cass, G. R. and Simoneit, B. R. T.: Sources of Fine Organic Aerosol. 3. Road Dust, Tire Debris, and Organometallic Brake Lining Dust: Roads as Sources and Sinks, Environ. Sci. Technol., 27(9), 1892–1904, doi:10.1021/es00046a019, 1993.

Schefuß, E., Ratmeyer, V., Stuut, J. B. W., Jansen, J. H. F. and Sinninghe Damsté, J. S.: Carbon isotope analyses of *n*-alkanes in dust from the lower atmosphere over the central eastern Atlantic, Geochim. Cosmochim. Acta, 67(10), 1757–1767, doi:10.1016/S0016-7037(02)01414-X, 2003.

Shrivastava, M., Fast, J., Easter, R., Gustafson, W., Zaveri, R., Jimenez, J., Saide, P., Hodzic, A.: Modeling organic aerosols in a megacity: Comparison of simple and complex representations of the volatility basis set approach, Atmos. Chem. Phys., 11, 6639–6662, doi:10.5194/acp-11-6639-2011, 2011.

Shrivastava, M., Berg, L. K., Fast, J. D., Easter, R. C., Laskin, A., Chapman, E. G., Gustafson, W. I., Liu, Y., and Berkowitz, C. M.: Modeling aerosols and their interactions with shallow cumuli during the 2007 CHAPS field study, J. Geophys. Res. Atmos., 118(3), 1343–1360, doi:10.1029/2012JD018218, 2013.

Takada, H., Onda, T. and Ogura, N.: Determination of Polycyciic Aromatic Hydrocarbons in Urban Street Dusts and Their Source Materials by Capillary Gas Chromatography, Environ. Sci. Technol., 24(8), 1179–1186, doi:10.1021/es00078a005, 1990.

Van Drooge, B. L. and Grimalt, J. O.: Particle size-resolved source apportionment of primary and secondary organic tracer compounds at urban and rural locations in Spain, Atmos. Chem. Phys., 15(13), 7735–7752, doi:10.5194/acp-15-7735-2015, 2015.

Zhao, Y., Nguyen, N. T., Presto, A. A., Hennigan, C. J., May, A. A. and Robinson, A. L.: Intermediate Volatility Organic Compound Emissions from On-Road Diesel Vehicles: Chemical Composition, Emission Factors, and Estimated Secondary Organic Aerosol Production, Environ. Sci. Technol., 49(19), 11516–11526, doi:10.1021/acs.est.5b02841, 2015.

Zheng, J., He, M., Shen, X., Yin, S. and Yuan, Z.: High resolution of black carbon and organic carbon emissions in the Pearl River Delta region, China, Sci. Total Environ., 438, 189–200, doi:10.1016/j.scitotenv.2012.08.068, 2012.

---

## Author Comment (AC2) · 14 May 2019

Response to the Reviewer We appreciate the reviewer for his/her constructive criticisms and valuable comments, which were of great help in improving the quality of the manuscript. We have revised the manuscript accordingly and our detailed responses are shown below. All the revision is highlighted in the revised manuscript.

Referee 1 Comments General comments: The authors developed a high resolution emission inventory of semi-volatile and intermediate volatility organic compounds (S/IVOCs) for the Pearl River Delta region, and then evaluated the impacts of anthropogenic S/IVOCs on secondary organic aerosols (SOA) by a regional chemical

transport model. The primary emissions and chemical degradation mechanisms of S/IVOCs are among the key knowledge gaps in better understanding and predicting the SOA formation. Thus this study is useful for future modelling studies of SOA in the PRD region. I would recommend that this manuscript can be considered for publication after the following specific comments being addressed. Reply: We thank the reviewer's positive comments and helpful suggestions. We have addressed all of the comments/suggestions in the revised manuscript. Detailed responses to the individual specific comment/suggestion are as follows.

Specific comments: R1.1. Introduction: a brief introduction of S/IVOCs at its first appearance should be helpful for the readership to better understand the context of this study. For example, what compounds do the S/IVOCs include? What are their major sources? Etc. Reply: Thanks for the helpful suggestion. We have added the following text to Section 1 to give a brief introduction of S/IVOCs in the revised manuscript: "[. . .] To date, S/IVOCs are found to mainly include straight chain and branched alkanes with carbon numbers > 12, alkylcyclohexanes, unsubstituted and substituted polycyclic aromatic hydrocarbons (PAHs), alkylbenzenes, cyclic and polycyclic aliphatic material (Zhao et al., 2015; Li et al., 2018; Drozd et al., 2019). However, a vast majority of S/IVOC mass still have not been speciated at the molecular level, which are defined as an unresolved complex mixture (UCM) (Jathar et al., 2012; Zhao et al., 2015; Drozd et al., 2019). Incomplete combustion, such as the combustion of fossil fuel, especially vehicle exhaust has been reported to be a large contributor to S/IVOC emissions in developed regions (May, Presto, et al., 2013a, 2013b; Ots et al., 2016; Khare and Gentner, 2018). Recent studies have also shown that consumer products and commercial or industrial products, processes, and materials are significant sources of unspeciated S/IVOCs (Czech et al., 2016; Khare and Gentner, 2018). On the other hand, biogenic S/IVOCs have recently been demonstrated to have a non-negligible impact on SOA formation, but very few measurements have been reported on their emissions (Palm et al., 2016, 2017). [. . .]" For details, please refer to Lines 7-19, Page 4 in the revised manuscript.

R1.2. Section 2.1: it is not clear if this S/IVOCs emission inventory only include one 'bulk species' or include individually many S/IVOC compounds for model use? I also wonder if there are biogenic sources of S/IVOCs. If so, the authors may need elaborate that this study mainly focused on anthropogenic emissions. Reply: The reviewer's comment is highly appreciated. In this study, we used source-specific linear scaling factors between emission factors of S/IVOCs and POA to establish the S/IVOCs emission inventory, where S/IVOCs has been lumped into a "bulk species" as suggested by previous studies (Hodzic et al., 2010; Shrivastava et al., 2011; Koo et al., 2014). The total S/IVOCs emission was not split into emission of the individual S/IVOC species because of the lack of accurate identification of S/IVOCs from different emissions (Jathar et al., 2012; Zhao et al., 2015; Drozd et al., 2019). Therefore, our S/IVOCs emission inventory treated S/IVOCs as one 'bulk species' for model use. On the other hand, even though previous studies have identified that biogenic sources could emit certain amounts of S/IVOCs, their contributions to SOA formation were insignificant in developed regions where anthropogenic emissions dominated (Palm et al., 2016, 2017; Khare and Gentner, 2018). To describe the emissions of S/IVOCs and their contributions to SOA formation more clearly, the text has been revised as followed: "[. . .] anthropogenic emissions and the chemical mechanisms of S/IVOCs have been incorporated into different models [. . .]" "[. . .] It should be noted that this study mainly focused on anthropogenic S/IVOCs and their roles in SOA formation in the PRD region as anthropogenic S/IVOCs were found to have much greater contributions to SOA formation than biogenic S/IVOCs in developed regions (Palm et al., 2016, 2017; Khare and Gentner, 2018)." For details, please refer to Lines 20-21, Page 4, Lines 26-27, Page 5 and Lines 1-2, Page 6 in the revised manuscript.

R1.3. Section 3.1: I am curious that dust is a source of S/IVOCs. It would be helpful if the authors elaborate more about this source by several sentences. Reply: Thanks for the reviewer's comment. In our emission inventory, dust mainly includes road fugitive dust and building construction dust. Particles containing many toxic metals and organic contaminants such as PAHs and long-chain alkanes from various sources (e.g.,

weathered materials of street surfaces, automobile exhaust, lubricating oils, gasoline, diesel fuel, tire particles, construction materials and atmospherically deposited materials), can be deposited on roads and construction sites, which are known as road fugitive dust and building construction dust (Takada et al., 1990; Rogge et al., 1993; Chen et al., 2012). Furthermore, dust is a large source of POA at urban locations, and S/IVOCs are frequently co-emitted with POA (Zheng et al., 2012; Shrivastava et al., 2013; van Drooge and Grimalt, 2015). Indeed, S/IVOCs, such as n-alkanes (C19–C39) and PAHs have been identified in dust samples, confirming that dust could be a source of S/IVOCs (Takada et al., 1990; Rogge et al., 1993; Schefuß et al., 2003; Dong and Lee, 2009). Therefore, it is reasonable that dust is a source of S/IVOCs. The above description for the fact that dust is a source of S/IVOCs has been provided in the revised manuscript. For details, please refer to Lines 1-11, Page 9 in the revised manuscript.

R1.4. P1, L13: change "the Pearl River Delta (PRD)" to "the PRD", as you have already defined the PRD in Line 12. Reply: Sorry for the mistake. It has been revised in the revised manuscript (Line 13, Page 1).

R1.5. P1, L15: emission factors of POA. . . Reply: Sorry for the mistake. It has been revised in the revised manuscript (Lines 14-15, Page 1).

R1.6. P5, L16-17: I suggest the authors to provide a brief description of the definition of parameters used in this Equation (1) here, so that the readers do not need to refer to the supplement. Reply: Thanks for the suggestion. The description of the definition of parameters used in Equation (1) has been moved from the supplement to the main text. For details, please refer to Lines 7-9, Page 6 in the revised manuscript.

R1.7. P9, L13-14: the same as the above comment. Reply: Thanks for the suggestion. The description of the definition of parameters used in Equation (4) and (5) has been removed from the supplement to the main text. For details, please refer to Lines 15-23, Page 10 in the revised manuscript.

R1.8. P10, L6: the roles of S/IVOCs in the formation of SOA. . . Reply: Sorry for the mistake. It has been revised in the revised manuscript (Line 13, Page 11).

R1.9. P15, L13: in my opinion, WQS should be a regional receptor site in the PRD region other than a regional background site, as it is generally located at the downwind of city clusters in winter monsoon season. Reply: We agreed with the reviewer that WQS is generally located at the downwind of city clusters of PRD in winter monsoon season. Indeed, in winter monsoon season, air masses could bring air pollutants from the center areas of PRD to the WQS site, making WQS site as a regional receptor site of PRD and it was frequently used to investigate the characteristics of air pollutants in PRD region (Guo et al., 2009; Ding et al., 2012). Therefore, the text has been revised as followed: "In this study, daily measured concentrations of SOA at the WQS site in Guangzhou, a receptor site of the PRD region during autumn and winter seasons, were used to evaluate the model performance on the simulation of SOA (Ding et al., 2012). The monitoring data of this site could represent the regional air pollution in the PRD [. . .]" For details, please refer to Lines 6-9, Page 17 in the revised manuscript.

R1.10. Table 3: I suggest moving this table to the supplement. Reply: Thanks for the reviewer's comment. Table 3 has been moved to the supplement accordingly.

R1.11. Is this newly developed emission inventory can be used by the modelling community? If so, how can it be accessed? A comment on this should be given in the data availability. Reply: Thanks for pointing this out, to provide the access for the developed emission inventory, a description has been added as followed: "The underlying research data and the newly developed emission inventory of S/IVOCs in this study are available to the community and can be accessed by request to Xuemei Wang (eciwxm@jnu.edu.cn) of Jinan University." For details, please refer to Lines 5-7, Page 22 in the revised manuscript.

Please also note the supplement to this comment:
https://www.atmos-chem-phys-discuss.net/acp-2018-1341/acp-2018-1341-AC2-

supplement.pdf